# RANDOMIZED SIGNATURE LAYERS
# FOR SIGNAL EXTRACTION IN TIME SERIES DATA

## ABSTRACT

Time series analysis is a widespread task in Natural Sciences, Social Sciences, and Engineering. A fundamental problem is finding an expressive yet efficient-to-compute representation of the input time series to use as a starting point to perform arbitrary downstream tasks. In this paper, we build upon recent works that use the Signature of a path as a feature map and investigate a computationally efficient technique to approximate these features based on linear random projections. We present several theoretical results to justify our approach and empirically validate that our random projections can effectively retrieve the underlying Signature of a path. We show the surprising performance of the proposed random features on several tasks, including (1) mapping the controls of stochastic differential equations to the corresponding solutions and (2) using the Randomized Signatures as time series representation for classification tasks.When compared to corresponding truncated Signature approaches, our Randomizes Signatures are more computationally efficient in high dimensions, and often lead to better accuracy and faster training. Besides providing a new tool to extract Signatures and further validating the high level of expressiveness of such features, we believe our results provide interesting conceptual links between several existing research areas, suggesting new intriguing directions for future investigations.

## 1 INTRODUCTION

Modeling time series is a common task in finance, physics, and engineering. A frequent challenge is finding a transformation mapping a newly observed time series into a target one (seq2seq modeling) or into a label summarizing its salient properties (classification). In the absence of any principled model describing such a mapping, one has to infer it from data. The last few years have witnessed the rise of deep neural networks, which have found successful application to problems involving time series in numerous domains (Fawaz et al., 2019; Gamboa, 2017). Nevertheless, their outstanding performance comes at the price of over-parametrization, data hungriness, and expensive training cost (Werbos, 1990; Brown et al., 2020; Teubert et al., 2021; Neyshabur et al., 2018; Marcus, 2018). Furthermore, even if sufficient data is available, the resulting models learn representations of the input data that are highly specialized to the training task and difficult to adapt in different contexts. In addition, the remarkable performance of these methods is often the result of a substantial engineering effort and is not supported by theoretical results.

*Reservoir Computing* (RC) (Schrauwen et al., 2007) offers an intriguing alternative strategy to cope with the limitations above, yet retaining the universal approximation properties typical of deep neural networks. In RC, the learning is divided into two phases: first, data is passed through an untrained reservoir which extracts a set of task-independent features; second, a simple and efficient-to-train linear map (the readout map) projects such features into the desired output. The critical point is that the design of the reservoir determines the expressiveness of the features, and several alternatives can be found in the literature (see (Gauthier et al., 2021) and references therein).

A powerful reservoir is offered by the *Signature Transform*, often simply referred to as Signature, stemming from rough path theory (Ben Hambly, 2010; Friz and Hairer, 2020). The Signature of a path consists in enhancing the path with additional curves, which, in the smooth case, corresponds to iterated integrals of the curve with itself. A profound mathematical result (Levin et al., 2013) guarantees that the solution of a (rough) differential equation can be approximated arbitrarily well

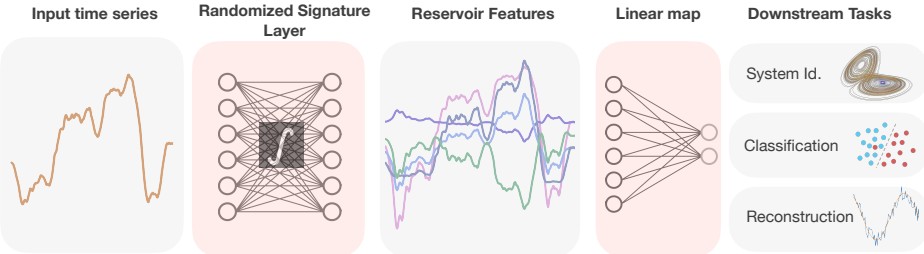

Figure 1: Reservoir computing via random Signature layers. Given an input time-series, our approach can be seen as random Signature layer, acting as a reservoir, extracting a set of random features preserving the strong theoretical properties of the Signature of the input path. A simple linear readout can the be applied to the so extracted features to perfrom arbitrarily downstream tasks.

by a *linear map* of the Signature of the input signals (a.k.a. controls). In the machine learning jargon, the Signature of a path can be interpreted as a feature map extracting all its geometrical properties and thus allowing a simple linear map to approximate any function of it. On the other hand, it is often the case that the reservoir features are very high dimensional, and hence are particularly expensive to calculate and use in downstream tasks. In addition, the high-dimensionality of the Signature reservoir poses additional challenges for modern gradient-based optimizers as convergence rates suffer from a linear dependence in the model dimension (Bottou et al., 2018).

Inspired by the remarkable theoretical properties of the Signature reservoir and motivated to fix its practical pitfalls, our contribution is to showcase the effectiveness of *Randomized Signatures* (Cuchiero et al., 2021b;a), a recently introduced reservoir of random features. These features provably hold the same geometric properties and approximation power as the Signature, yet are often more *efficient to compute* and of lower dimensionality. To extract Randomized Signatures, we numerically integrate a set of random linear stochastic differential equations driven by the path itself. We assess the expressiveness of the proposed random features on several tasks, including non-parametric (black-box) system identification problems arising from complex controlled nonlinear dynamical systems as well as time series classification.

The paper is structured as follows: In Section 2, we link the Randomized Signature approach with related research areas in the literature; in Section 3, we present relevant theoretical results motivating our method; in Section 4, we assess the effectiveness of Randomized Signatures on various system identification and time series classification tasks, showing promising results when compared to other signature approaches for reservoir computing. Finally, in Section 5, we discuss our conclusions and possible future developments.

## 2 RELATED WORKS

**Random Features and Reservoir Computing.**   The idea of extracting features based on random operations is not new and has seen a number of successful applications over the past years. Of particular note, the seminal work of Rahimi and Recht (2008) proposes to accelerate kernel machines by learning random features whose inner product matches that of a target shift-invariant kernel. The trade-off between generalization and computational efficiency of learning with random features has then been rigorously studied by Rudi and Rosasco (2017). A conceptually very similar rationale is introduced by a parallel series of works exploring the topic of Reservoir Computing (Schrauwen et al., 2007). Similarly to us, Echo State Networks (Jaeger, 2003) evolve the input state by a series of fixed random projection (the reservoir) and generate the output by applying a trainable linear projection over the hidden states. However, we make the additional step of linking the random features to the Signature of the input path and, as shown below, the evolution of the features is dictated by a randomly-evolved stochastic differential equation driven by the input path.

**Controlled Differential Equations.**   Our work is also related with a series of recent papers investigating the problems of how to process irregular time series and to condition a model on incoming information through the lens of controlled differential equations (CDEs) (Kidger et al., 2020; Morrill et al., 2021). Interestingly, Kidger et al. (2020) show that the action of a linear layer on the final

output of a trained Neural CDEs – the extension of Neural ODEs (Chen et al., 2018) to CDEs – results in a universal function approximator. Differently from them, our method is less computationally expensive as the only parameters we need to train are those of the final linear readout layer. Fundamentally different to our work, the approach of Morrill et al. (2020) first randomly projects the possibly high dimensional controls into a lower dimensional space and then extracts the Truncated Signature from such compressed input. Our method instead extracts a random compression of the signature of the original input controls.

**Rough Path Theory.** Rough path theory is about describing how possibly highly oscillatory (rough) control path interact with nonlinear systems (Lyons, 2014). The concept of Signature is introduced in this context to provide a powerful description of the input path, removing the need to look at the fine structure of the path itself. Recent years have seen a resurgence of these ideas, which have been revisited from a machine learning perspective (Bonnier et al., 2019; Kidger and Lyons, 2021). Our analysis is strongly influenced by the work of Cuchiero et al. (2021a;b), who, starting from the observation that the Signature is an infinite dimensional reservoir, establishes that its information content can be efficiently compressed by a random projection performed by a dynamical system driven by random vector fields. As we show later, this results in a *random* reservoir which preserves the properties of the Signature while living in a finite dimensional space.

## 3 BACKGROUND

We provide the theoretical tools supporting the Randomized Signature approach for feature extraction in times series. A formal discussion can be found in (Friz and Hairer, 2020) and (Cuchiero et al., 2021b).

### 3.1 RANDOMIZED SIGNATURE OF A PATH

Let $X : [0, T] \to \mathbb{R}^d$ be a continuous piecewise smooth $d$-dimensional path $X = (X^1, \cdots, X^d)$. We will refer to $X$ as the *control* and to its single components $X_i$ as *controls*. We denote by $\{e_1, \ldots, e_d\}$ the canonical basis of $\mathbb{R}^d$.

**Definition 1 (Signature)** *For any $t \in [0, T]$, the Signature of a continuous piecewise smooth path $X : [0, T] \to \mathbb{R}^d$ on $[0, t]$ is the countable collection $\mathbf{S}_t := (1, S_t^1, S_t^2, \ldots) \in \prod_{k=0}^{\infty} (\mathbb{R}^d)^{\otimes k}$ where, for each $k \geq 1$, the entries $S_t^k$ are the iterated integrals defined as*

$$S_t^k := \sum_{(i_1,\ldots,i_k) \in \{1,\ldots,d\}^k} \left( \int_{0 \leq s_1 \leq \cdots \leq s_k \leq t} dX_{s_1}^{i_1} \ldots dX_{s_k}^{i_k} \right) e_{i_1} \otimes \cdots \otimes e_{i_k}. \tag{1}$$

*We define the Truncated Signature of $X$ of order $M \geq 0$ as*

$$\mathbf{S}_t^M := (1, S_t^1, \ldots, S_t^M) \in \prod_{k=0}^{M} (\mathbb{R}^d)^{\otimes k} =: \mathcal{T}^M (\mathbb{R}^d). \tag{2}$$

A practical example of computation of Signatures is presented in Appendix A.3.

The definition in the last paragraph suggests that the Signatures of $X$ can be used to approximate any regular enough function of $X$, for instance the solution to differential equations controlled by $X$. The following result makes this argument precise in the multidimensional setting.

**Theorem 1 (Signature is a Reservoir)** *Let $V_i : \mathbb{R}^m \to \mathbb{R}^m, i = 1, \ldots, d$ be vector fields regular enough s.t. $dY_t = \sum_{i=1}^{d} V^i (Y_t) dX_t^i, Y_0 = y \in \mathbb{R}^m$, admits a unique solution $Y_t : [0, T] \to \mathbb{R}^m$. Then, for any smooth test function $F : \mathbb{R}^m \to \mathbb{R}$ and for every $M \geq 0$ there is a time-homogeneous linear operator $L : \mathcal{T}^M (\mathbb{R}^d) \to \mathbb{R}$ which depends only on $(V_1, \ldots, V_d, F, M, y)$ s.t.*

$$F (Y_t) = L (\mathbf{S}_t^M) + \mathcal{O} (t^{M+1}), \quad t \in [0, T]. \tag{3}$$

This theorem suggests the first $M$ entries of the Signature of $X$ are sufficient to linearly explain the solution of any differential equation driven by it. In addition to this, it supports the claim that such features are valuable for any downstream usage.

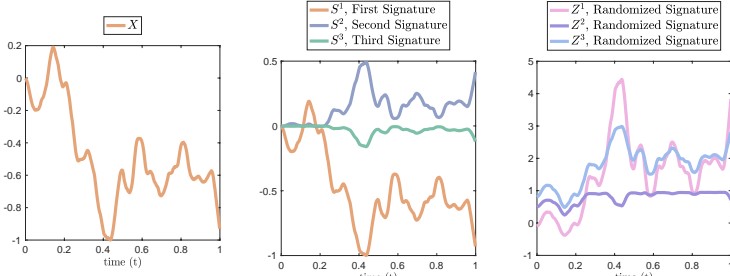

Figure 2: To enhance intuition, we show the Signatures and Randomized Signatures for $X$ (smoothed Brownian Motion). Note that $X_t = S_t^1 + X_0$ for each $t$, and that $S^2$ and $S^3$ get smaller and smaller in magnitude, yet more and more regular (smoother, as a result of iterated integrals). The Randomized Signatures resemble a mix (i.e. a random linear combination of) the true Signatures: $Z^1$ resembles a shifted version of $-S^1$, while $Z^2$ resembles a combination of $S^1$ and $S^3$. Indeed, Thm. 2 guarantees that Randomized Signatures approximate the full Signature with no need for computation of iterative integrals. We validate this experimentally in Sec. 4.

## 3.2 RANDOMIZED SIGNATURE OF A PATH AND ITS THEORETICAL GUARANTEES

Calculating $\mathbf{S}_t^M$ requires the calculation of $\frac{d^{M+1}-1}{d-1}$ iterated integrals (Appendix A.3) – which in total quickly becomes computationally expensive. Several computational techniques have been developed to circumvent this problem, see, e.g. (Kidger and Lyons, 2021). The next result provides a practical description of how it is possible to *reduce the computational burden without losing too much explanatory power*. As such, this results provides the theoretical foundation for our approach.

**Theorem 2 (Randomized Signature (Informal))** *For any $k \in \mathbb{N}$ big enough and appropriately chosen random matrices $A_1, \ldots, A_d$ in $\mathbb{R}^{k \times k}$, random shifts $b_1, \ldots, b_d$ in $\mathbb{R}^{k \times 1}$, random starting point $z$ in $\mathbb{R}^{k \times 1}$, and any fixed activation function $\sigma$, the solution of*

$$dZ_t = \sum_{i=1}^{d} \sigma\left(A_i Z_t + b_i\right) dX_t^i, \quad Z_0 = z \in \mathbb{R}^k, \quad t \in [0, T]. \tag{4}$$

*– called the Randomized Signature of $X$ – **has comparable approximation power as the Signature itself and maintains its geometric properties.***

The formal statement of such a result is the combination of Theorem 8, Theorem 9, and Definition 10 (see Appendix). A complete formal discussion as well as the original statement of this theorem can be found in the paper (Cuchiero et al., 2021a). In a nutshell, the Randomized Signature $Z$ follows a dynamics which provides an efficient and powerful compression of the Signature through a low-dimensional random projection. The expressiveness of such projection is guaranteed by a standard Johnson-Lindenstrauss argument (Cuchiero et al., 2021a).

**Computational complexity and dimensionality of Randomized Signatures.** The computational complexity of calculating $Z_t$ is $\mathcal{O}(k^2 d)$, while its dimensionality is $\mathcal{O}(k)$. However, we should then ask the crucial question "*how should we compare $k$ with $M$?*". We show experimentally in Section 4 that, in order to match the approximation guarantees of the Truncated Signature of order $M$, the number of required Randomized Signatures $k$ is not too big — in particular it is not exponential in $M$. This confirms that working with Randomized Signatures is often less computationally demanding and results in lower-dimensional features.

We conclude the section remarking the fascinating implications of choosing a random $A_i$s and $b_i$s.

**Theorem 3 (Density of Randomized Signatures (Informal))** *For any sequences of time points $0 \leq t_0 < \ldots < t_N \leq T$ and points $z_{t_0}, \ldots, z_{t_N} \in \mathbb{R}^k$ we can find a smooth control $X$ such that the solution $Z$ of Equation 4 at time $t_i$ is such that $Z_{t_i} = z_{t_i}$, for $i = 0, \ldots, N$. Additionally, if the control $X$ is a $d$-dimensional Brownian motion, then the solution of Equation 4 at any point in time $t > 0$ admits a smooth density with respect to Lebesgue measure.*

## 4 THE EXPRESSIVE POWER OF RANDOMIZED SIGNATURES

To compute the Randomized Signature, we rely on an Euler-Maruyama approximation; details are given in Algorithm 1. To start, we experimentally validate some of the results presented above. In

---

**Algorithm 1** Generate Randomized Signature

---

**Require:** $X \in \mathbb{R}^d$ sampled at $0 = t_0 < \cdots < t_N = T$, Randomized Signature dimension $k$, activation function $\sigma$.
   Initialize: $Z_0 \in \mathbb{R}^k, A_i \in \mathbb{R}^{k \times k}, b_i \in \mathbb{R}^k$ to have iid standard normal entries
   **for** $n$ in $\{1..N\}$ **do**
$$Z_{t_n} = Z_{t_{n-1}} + \sum_{i=1}^d \sigma\left(A_i Z_{t_{n-1}} + b_i\right)\left(X_{t_n}^i - X_{t_{n-1}}^i\right)$$
   **end for**

---

particular, we show that the Truncated Signature of a control lies in the linear span of its Randomized Signatures, thus validating Theorem 2. As a second proof of concept, we reconstruct a control $X$ as a linear combination of its Randomized Signatures, *showcasing the expressiveness of such random features*. Finally, we combine Theorem 1 and Theorem 2 and we use $Z$ to learn the dynamics of complex nonlinear systems of differential equations and to perform time series classification. Our result show that the Randomized Signature is a powerful random reservoir which guarantees — both in theory and practice — the same geometrical properties and approximation power as the Signature itself, while often being of lower dimensionality and less computationally expensive both at extraction and at deployment time.

**Note on the choice of $\sigma$.** While the choice of $\sigma$ does not affect the theoretical results (Cuchiero et al., 2021a;b), we found that a careful choice is needed for optimal results. Inspired by seminal works on the stability of deep linear networks (Glorot and Bengio, 2010) and by the connection to neural ODEs (Chen et al., 2018), we selected $\sigma$ to be linear, with slope $\frac{1}{d \times \sqrt{k}}$. This choice, complemented with additional fine-tuning, leads to the best performance for downstream tasks.

## 4.1 LEARNING THE SIGNATURES

In this experiment, we show that we can approximate the Truncated Signature of a control using a linear combination of its Randomized Signatures. We fix an equally-spaced partition $\mathcal{D} = \{t_0, \cdots, t_N\}$ of $[0, 1]$ and calculate the truncated Signature of order $M$ of a $d$-dimensional path $X$ on $\mathcal{D}$, i.e. $\mathbf{S}_{t_i}^M$, for $i = 0, \cdots, N$. For a fixed value of $k$, we use Algorithm 1 to generate a realization of the Randomized Signature $Z$ of $X$ on $\mathcal{D}$. For sake of clarity, the truncated Signatures on $\mathcal{D}$ are reshaped into an $\left(N \times \left(\left(d^{(M+1)} - 1\right)/(d-1) - 1\right)\right)$-dimensional matrix called $S_M^{true}$ while the Randomized Signatures on $\mathcal{D}$ is represented as an $(N \times k)$-dimensional matrix called $Z_k^{Rand}$. Finally, we train a Ridge Regression to find a parameter matrix $W \in \mathbb{R}^{k \times \left(\left(d^{(M+1)}-1\right)/(d-1)-1\right)}$ mapping the Randomized Signatures $Z_k^{Rand}$ into the truncated Signatures $S_M^{true}$.

More precisely, we consider a 10-dimensional control $X = (t, \mathbf{W}_t)$ where $\mathbf{W}$ is a 9-dimensional Brownian motion with independent components, and fix $M = 6$. For each $k \in \{1, \cdots, 200\}$, we generate $Z_k^{Rand}$, perform a simple linear regression and calculate the $L^2$ approximation error. Results in Figure 3 indicate that the approximation error decreases as $k$ increases. This confirms that, as ensured by Theorem 2, the random features constructed as prescribed by Equation 4 hold the same information content as the Signature and thus maintain its approximation power and geometric properties. Finally, we highlight that, if we were to be satisfied with an approximation error of order $10^{-4}$, we could achieve it with roughly with $k = 190$ Randomized Signatures. To conclude, this means that, instead of calculating $\left(\left(d^{(M+1)} - 1\right)/(d-1) - 1\right) = 1111110$ integrals per time step, we could just perform $k^2 d = 36100$ calculations per time step — *which is 3 times cheaper*. More remarkably, this high feature dimensionality (1111110 for Signature) makes any model for downstream tasks exponentially more complex than the that based on Randomized Signature (only 190 features). Modeling choices and details are reported in Appendix B.2.

## 4.2 RECONSTRUCTION AND RANDOMIZED SIGNATURE AUTOENCODERS

The Randomized Signature of $X$ is rich enough to reconstruct it almost perfectly. In practice, we consider an "autoencoder" architecture, whose encoder is our random feature generator and the decoder is a trained linear projection mapping the Randomized Signatures into the original path. For this experiment, we fix an equally-spaced partition $\mathcal{D} = \{t_0, \cdots, t_N\}$ and consider a single

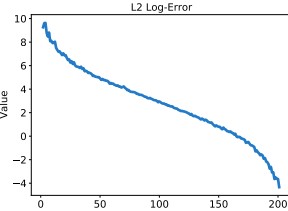 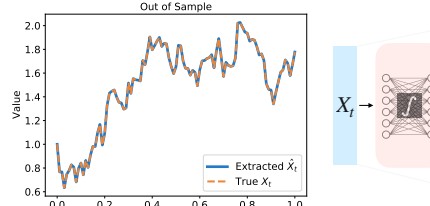

Figure 3: Randomized Signature approximates the Signature as $k$ increases.

Figure 4: Performance of the Randomized Signature autoencoder.

realization $X_{Train}$ of an Ornstein–Uhlenbeck process. Ultimately, we train a Ridge Regressor to map the Randomized Signature $Z$ of the one and only train sample $X_{Train}$ into $X_{Train}$ itself. Table 6 in Appendix shows the average reconstruction error on $N_{Test} = 10000$ test samples as we repeat the experiment for different choices of $k$. Such results are in accordance with Figure 4 which shows an example of the true and generated time series of a test sample. This confirms that the Randomized Signature of a path holds an equivalent information content as the path itself while being more usable for any downstream tasks. Modeling choices and details are reported in Appendix B.2.

### 4.3 LEARNING SDE SOLUTIONS WITH A LINEAR MAP ON THE RANDOMIZED SIGNATURE

In this series of experiments, we provide evidence of the potential of Randomized Signatures in several downstream tasks. First of all, as shown in Appendix B, we prove that our approach can accurately learn the dynamics of several rough differential equations: we highlight that the algorithm has only access to the input controls and the output response of the system but it is *completely agnostic with respect to the law of the dynamics itself.* To start with, we focus on the nonlinear dynamics of 1-dimensional stochastic Double Well. Secondly, we study the multidimensional case of the 4-dimensional Ornstein–Uhlenbeck. Then, we investigate the performance on the extremely rough dynamics of Fractional Geometric Brownian Motion even for low values of the Hurst parameter $H$. Additionally, we obtain promising results in learning the dynamics of an Enzyme-Substrate Reactions (Ingalls, 2013). We also demonstrate that this approach has sustainable performance even when the input control is irregularly sampled as well as Neutral Controlled Differential Equations. Due to space constraints, these results are presented in the Appendix. Here, we show the performance in learning highly nonlinear dynamical systems with a real world application.

**Tumor Growth Kinetics.** In this experiment, we consider predicting the output of a complex controlled nonlinear system using Randomized Signatures. The Simeoni Tumor model (Simeoni et al., 2004) is widely used to predict the evolution of tumor weight $Y_t$ under the concentration of a treatment drug $X_t$. The system is specified by the following equations

$$du_t^1 = \left\{ \lambda_0 u_t^1 \left[ 1 + (\lambda_0/\lambda_1 w_t)^\Psi \right]^{-1/\Psi} - k_2 X_t u_t^1 \right\} dt, \quad du_t^2 = \left[ k_2 X_t u_t^1 - k_1 u_t^2 \right] dt,$$

$$du_t^3 = \left[ k_1 \left( u_t^2 - u_t^3 \right) \right] dt, \quad du_t^4 = \left[ k_1 \left( u_t^3 - u_t^4 \right) \right] dt, \quad Y_t = u_t^1 + u_t^2 + u_t^3 + u_t^4. \tag{5}$$

We fix an equally spaced partition $\mathcal{D}$ of $[0, 10]$ and perform Ridge Regression to map 10000 instances of the Randomized Signature of the controls into the respective integrated solution $Y_t$. We sample $X_t$ from the law of a Scaled Squared Brownian motion $0.25 \cdot W_t^2$ where $W_t$ is a 1-dimensional Brownian Motion (BM) and the left side of Figure 5 shows the comparison of the true and the generated time series on a test sample along with the respective path of the control. To test the robustness of our approach, we use controls that do *not follow the law used in training*. More precisely, the middle of Fig. 5 shows what happens if we use $\mathbb{1}_{\{W_t^2 > 1\}}$ as input control instead of $0.25 \cdot W_t^2$ while its right does the same as we use $W_t$. These two experiments prove that the algorithm learnt much more than merely predicting the behaviour of the system under ordinary circumstances. The first shows that the algorithm has "understood" that the absence of drug leads to the growth of the tumor while it dies back once the medicine is injected. On the other hand, the second suggests that the algorithm cannot behave properly when the input becomes negative — not allowed by the problem definition (negative drug concentration). Further details are reported in Appendix B.2.

**Electrochemical Battery Model.** In this experiment, we learn the dynamics of the electrochemical battery model proposed in (Daigle and Kulkarni, 2013) which returns the voltage $Y$ as the current

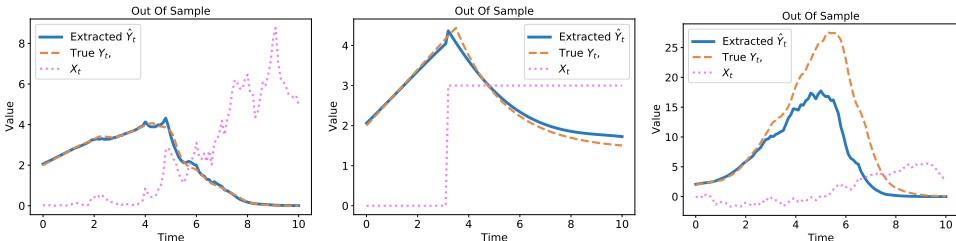

Figure 5: Tumor Growth Kinetics stimulated with Scaled Squared BM (left) - Step Function (center) - BM (right). Model is only trained on scaled squared BMs, so the input in the central plot is *out of distribution*.

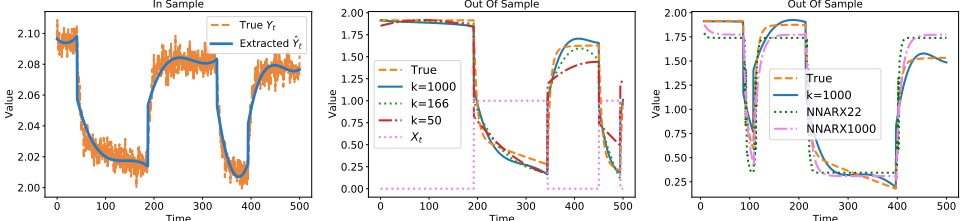

Figure 6: Electrochemical Battery Model: Example of fit for a Train Sample (left) - Predictions on Test Sample for different $k$ (left) - Comparison with NNARX (right)

$X$ is drawn from the battery. We use the open source NASA Prognostic Model Package (Teubert et al., 2021) to simulate voltage trajectories given input current control paths. On a fixed equally spaced partition $\mathcal{D}$ of $[0, 500]$, we model the input current with step functions taking values 0 or 1 on random sub-interval of $[0, 500]$. To test the robustness of our approach, we train a Ridge Regression to map $N_{Train} = 1000$ instances of $k$-dimensional Randomized Signature of the controls into the respective solutions to which we add white noise. The left of Figure 6 shows the comparison between the ground truth and our prediction on an in-sample trajectory. In the same figure, we compare the generated time series for several choices of $k$ on a test sample along with the path of the input control. In Figure 6 we show the performance of our method in comparison with a nonlinear neural ARX (NNARX) — which is often used in the modern control theory/system identification literature as a benchmark (see e.g. (Masti and Bemporad, 2018)), given its strong theoretical guarantees (Schoukens and Ljung, 2019). Table 8 in Appendix reports the performances in terms of MSE on 1000 test samples for our models compared to two versions of NNARX: NNARX 22 which has around 1000 trainable parameters just like our model with $k = 1000$, and NNARX 1000 which matches our best model in terms of MSE but has around $10^6$ trainable parameters. Details are reported in Appendix B.2.

**Comparison with Neural Controlled Differential Equations.** In Appendix B, we benchmark the performance of our model with that of NCDEs (Kidger et al., 2020), a recently proposed competitive deep learning technique for controlled dynamical system modelling. We test the two models on the task of learning the dynamics of an Ornstein-Uhlenbeck Process, and showcase the superior performance of our method both in terms of computational complexity and accuracy.

## 4.4 COMPARISON WITH TRUNCATED SIGNATURES

In this experiment, we learn the dynamics of the Fractional Ornstein–Uhlenbeck process using our Randomized Signature model and benchmark our results against a Truncated Signature model of order $M$. The computational cost of extracting the Truncated Signature of order $M = 3$ for $d$ controls is $\mathcal{O}(d^3)$ while for Randomized Signature it is $\mathcal{O}(k^2 d)$. To keep the computational cost of extracting features equal across models, we select $k = d$. As a result, the number of features of the Signature-based model is $\mathcal{O}(d^3)$ while that of our model is $\mathcal{O}(d)$. In the spirit of Reservoir Computing, we choose both models to be linear. Hence, the number of trainable parameters is equal to the number of features. This implies that, the number of trainable parameters of our model is orders of magnitude smaller than that of our benchmark (left of Figure 7) — *this severely affects the training time (see e.g. (Bottou et al., 2018))*. Figure 7 shows that our training time is almost independent on $d$. Instead, the training complexity of the Signature-model is exponential in $d$ and

the Out-of-Sample performance degenerates as the underlying optimization problem explodes in dimension. This result clearly highlights the data-hungriness of Signature-based models when the number of dimensions is high. To conclude, we emphasize that if $d = 80$, the storage cost of the Truncated Signature of a path is 200MB while its Randomized Signature is just 32KB. As a consequence, training the former on $N_{Train} = 10000$ would take an unpractical amount of time and storage memory while we were able to train ours in $4.8896$ seconds using just 320MB and attaining an Out-of-Sample Error on $N_{Test} = 10000$ equal to $0.075329$. Modeling choices and details are reported in Appendix B.2

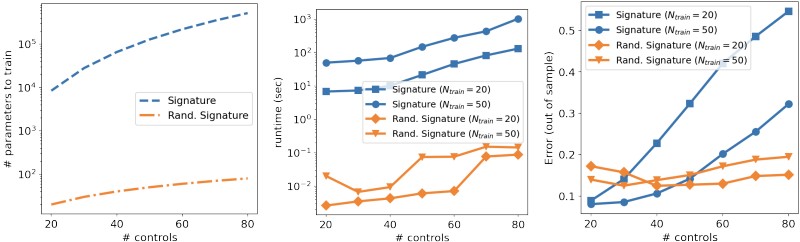

Figure 7: Randomized Signature vs. Truncated Signature model. (Left) Number of trainable parameters for Randomized Signature is significantly smaller regardless of the number of controls. (Center) Truncated Signature is much slower than Randomized Signature in high dimensions. (Right) As opposed to Randomized Signature, the performance of Truncated Signature degrades as the number of controls increases.

## 4.5 TIME SERIES CLASSIFICATION

In time series classification (TSC) the goal is to map an input time series to the corresponding label. Given the complex high-dimensional nature of temporal signals, the vast majority of classification algorithms aims at effectively finding expressive representations of the original time series and then maps them into the final prediction. This last operation is often realized via a relatively simple mapping, e.g. a linear map or a shallow neural network. Deep learning algorithms used for TSC include various types of deep convolutional neural networks which extract a hierarchical representation of the input time series by learning the parameters of their filters via backpropgation. These features are finally mapped into the logits associated with each class via a trainable linear projection. Last but not least, it is key to remark that these networks are typically over-parametrized and their training can be computationally demanding as well as challenging from an engineering perspective.

In this section, we investigate how we can actually replace deep neural networks used for feature extraction with relatively cheap random layers which iteratively generate a reservoir. Among other notable works, Rocket (Dempster et al., 2020) arguably proposes the most successful solution.

**Rocket.** This algorithm prescribes the generation of $k \gg 1$ unidimensional filters characterized by randomly drawn lengths, weights, and jumps. Given a filter of length $l_{filter}$, weights $\omega \in \mathbb{R}^{l_{filter}}$, and jump j, it can be easily shown that the feature $Z^i$ resulting from its action on an input path $X$, from position $i$ in $X$, evolves according to the following iterative update:

$$Z_k^i = Z_{k-1}^i + X_{i+(k\times j)} \times \omega_k, \quad k \in \{0, \cdots, l_{filter} - 1\} \tag{6}$$

where $Z_0^i = b$ and $b$ is a random real number. We define $Z^i$ to be equal to $Z_{l_{filter}-1}^i$. Depending on the values of j and $l_{filter}$ and on the length $l_X$ of the path $X$, this update produces a vector $\mathbf{Z} = (Z^1, \cdots, Z^M)$ of $M := l_X - (l - 1) \cdot j$ random features, used to extract only two nonlinear features: $max(\mathbf{Z})$ and $\frac{1}{M} \sum_{i=1}^M \mathbb{1}_{\{Z^i > 0\}}$. Ultimately, this means that if we use $k$ kernels, we extract $2k$ features per time series which are then used as input to a Ridge classifier to obtain the final prediction. Rocket results in state-of-the-art performances in TSC both in terms of accuracy and computational efficiency, outperforming several deep learning baselines which often involve order of magnitude more parameters to optimize.

**An alternative interpretation.** Looking at Eq. 6, it is natural to wonder if it can be replaced with an alternative and possibly more principled update scheme driven by the input path and evolved by random projections. With the intent of aligning Rocket to the framework presented in this paper, we replace Eq. 6 with the update equation in Algorithm 1, thus exploiting the expressive power of Randomized Signatures. As detailed in the Appendix B.2, the current implementation of this

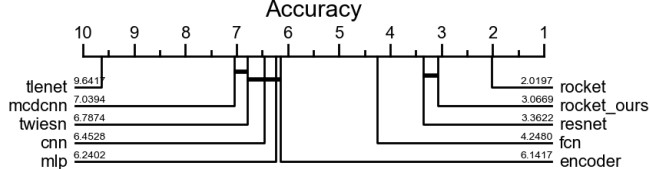

Figure 8: Mean rank of the modified version of Rocket described in the text and a number of deep learning baselines (results taken from (Fawaz et al., 2019)) on 128 datasets from the UCR archive. Our method is referred to as "rocket_ours". The other methods are, from left to right: Time Le-Net (Le Guennec et al., 2016), Multi Channel Deep Convolutional Neural Network (Zheng et al., 2014), Time Warping Invariant Echo State Network (Tanisaro and Heidemann, 2016), Time-CNN (Zhao et al., 2017), Multi Layer Perceptron (Wang et al., 2017), Encoder (Serrà et al., 2018), Fully Convolutional Neural Network (Wang et al., 2017), Residual Network (Wang et al., 2017).

algorithm is equivalent to assuming that the random vector field $A$ is approximately diagonal and autonomous. In other words, the Randomized Signatures are never mixed during their evolution and, differently from the update in Eq. 6, the filter weights do not depend on time. These represent two strong limitations which pave the way to several enhancements in future work. Despite these drastic shortcomings, the resulting algorithm outperforms several deep learning baselines on 128 TSC datasets from the UCR archive, as shown in Fig. 8. The original algorithm using Eq. 6 provides better results than ours but does not enjoy the same theoretical guarantees based on the theory of Randomized Signatures presented in Section 3.

## 5 CONCLUSIONS

**Reservoir Computing and the Concept of Signature.** The concept of reservoir computing is based on an simple and intriguing idea: generating a large number of features and then training a linear readout to perform an arbitrary downstream task. This simple approach promises to reduce the number of model parameters yet retaining a high level of expressive power. Inspired by results from rough path theory showing that the Signature Transform enjoys the status of reservoir, we propose a random reservoir, i.e. the Randomized Signature, to approximate the Signature Transform and thus inheriting its geometrical properties and expressive power. Following rigorous theoretical arguments, we do so by evolving a latent differential equation controlled by a path and evolved according to random vector fields. The empirical investigations presented in this paper provide evidence supporting that this strategy is successful and can be deployed on several tasks including non-parametric system identification and TSC.

**Future research and open questions.** From a theoretical standpoint, Eq. 4 is written in terms of autonomous vector fields $A_i$ for $i = 1, \cdots, d$, whose entries are drawn according to Gaussian distributions with zero mean and unit variance, and an activation function $\sigma$. It is natural to wonder if choosing time-dependent matrices $A_i$ leads to increased expressive power and to what extent the quality of the resulting features depend on a particular choice of $\sigma$. Furthermore, since our experiments suggest that the dimensionality $k$ of the random features play a significant role in terms of expressive power, it is not trivial to identify a rigorous criterion to determine it. A precise characterization of this quantity and a clever way to select its value could potentially lead to further savings in terms of computational cost.

From a practical point of view, our method proved to be able to accurately predict the response of different dynamical systems driven by heterogeneous and possibly irregular control paths, even when such controls were sampled from different laws with respect to the one used in training. In light of these results, it would be interesting to explore whether the method of Randomized Signatures could be extended to model the response of systems governed by *controlled partial differential equations*. In the context of TSC, we established a link with existing works using time series representations based on random features and our approach based on the update in Eq. 4. We implemented a first baseline algorithm combining our Randomized Signatures with the algorithm proposed in (Dempster et al., 2020), resulting in superior performance compared to strong deep learning baselines but sub-optimal results compared to the original algorithm. We believe such a gap can be filled by addressing the limitation of the current version of the algorithm, e.g. the assumptions it makes on the vector fields, as discussed at the end of Section 5. As a final consideration, we believe an interesting research direction being to investigate whether our random feature extraction pipeline could be incorporated into hierarchical models, such as deep neural networks, to form *hybrid methods* including both random and learned components operating in symbiosis.

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

# Appendix

## A  THEORY DETAILS

### A.1  BASIC DEFINITIONS

First of all, we define the concept of admissible tensor norms, which we assume to have in this work.

**Definition 2 (Admissible Tensor Norms)** *Let $E := \mathbb{R}^d$ and $\otimes$ be a tensor product such that the tensor powers of $E$, $\left( E^{\otimes k} : k \geq 1 \right)$, are equipped with a family $(\| \cdot \|_{E^{\otimes k}} : k \geq 1)$ of norms satisfying:*

1. *For $j, k \in \mathbb{N}$ and for all $h \in E^{\otimes j}$ and $l \in E^{\otimes k}$*

$$\|h \otimes l\|_{E^{\otimes(j+k)}} \leq \|h\|_{E^{\otimes j}} \|l\|_{E^{\otimes k}};$$

2. *For any permutation $\sigma$ of $\{1, \ldots, k\}$,*

$$\|l_1 \otimes \ldots \otimes l_k\|_{E^{\otimes k}} = \left\| l_{\sigma(1)} \otimes \ldots \otimes l_{\sigma(k)} \right\|_{E^{\otimes k}};$$

3. *For any bounded linear functionals $f$ on $E^{\otimes j}$ and $g$ on $E^{\otimes k}$, there exists a unique bounded linear functional, denoted as $f \otimes g$, on $E^{\otimes(j+k)}$ such that for all $h \in V^{\otimes j}$ and $l \in E^{\otimes k}$*

$$f \otimes g(h \otimes l) = f(h)g(l).$$

*A family of tensor norms satisfying these conditions is called a family of admissible tensor norms.*

Then we define the space in which the Signature of a path lies in, that is the following Tensor Algebra.

**Definition 3 (Tensor Algebra $\mathcal{T}\left( \mathbb{R}^d \right)$)** *We define the tensor algebra on $\mathbb{R}^d$ as*

$$\mathcal{T}\left( \mathbb{R}^d \right) := \prod_{k=0}^{\infty} \left( \mathbb{R}^d \right)^{\otimes k}$$

*as well as its truncated version of order $M \geq 0$ as*

$$\mathcal{T}^M\left( \mathbb{R}^d \right) := \prod_{k=0}^{M} \left( \mathbb{R}^d \right)^{\otimes k},$$

*where $\left( \mathbb{R}^d \right)^{\otimes k}$ is the space of tensors of shape $(d, \ldots, d)$ given by $\mathbb{R}^d \otimes \cdots \otimes \mathbb{R}^d$ for k times.*

**Definition 4 (Concatenation Operation)** *We define the Concatenation Operation $*$ such that for any given couple of continuous piecewise smooth paths $X : [0, s] \to \mathbb{R}^d$ and $Y : [s, t] \to \mathbb{R}^d$, their image through $*$ is the continuous piecewise smooth path $X * Y : [0, t] \to \mathbb{R}^d$ defined by*

$$X * Y_u := \begin{cases} X_u & \text{if } u \in [0, s] \\ X_s + Y_u - Y_s & \text{if } u \in [s, t] \end{cases}$$

**Definition 5 (Inverse Operation)** *We define the Inverse Operation $\leftarrow$ such that for any continuous piecewise smooth path $X : t \to \mathbb{R}^d$, its image through $\leftarrow$ is the continuous piecewise smooth path $\overleftarrow{X}_t := X_{T-t}$, for each $t \in [0, T]$.*

**Definition 6 (Topological Space)** *A topological space is an ordered pair $(X, \tau)$, where $X$ is a set and $\tau$ is a collection of subsets of $X$, satisfying the following axioms:*

1. *The empty set and $X$ itself belong to $\tau$;*

2. *Any arbitrary (finite or infinite) union of members of $\tau$ still belongs to $\tau$;*

3. *The intersection of any finite number of members of $\tau$ still belongs to $\tau$.*

*The elements of $\tau$ are called open sets and the collection $\tau$ is called a topology on $X$.*

**Definition 7 (Arcwise Connected Topological Space)** *A topological space $(X, \tau)$ is said to be arcwise connected if any two distinct points $x, y \in X$ can be joined by an arc, that is a continuous map $\alpha : [0, 1] \to X$ such that $\alpha(0) = x$ and $\alpha(1) = y$.*
*We say that $X$ is uniquely arcwise connected if for any two distinct points $x, y \in X$, there exists a unique path in $X$ that joins them.*

**Definition 8 ($\mathbb{R}$-tree)** *An $\mathbb{R}$-tree is a uniquely arcwise connected metric space, in which the arc between two points is isometric to an interval.*

**Definition 9 (Tree-Like)** *A continuous piecewise smooth path $x : [0, T] \to \mathbb{R}^d$ is tree-like if there exists an $\mathbb{R}$-tree $\tau$, a continuous map $\phi : [0, T] \to \tau$ and a map $\psi : \tau \to V$ such that $\phi(0) = \phi(T)$ and $x = \psi \circ \phi$.*

## A.2 SUPPORTING RESULTS

**Lemma 4 (Johnson-Lindenstrauss Lemma, (Cuchiero et al., 2021a))** *Given an $M$-dimensional Hilbert space $(H, \langle \cdot, \cdot \rangle_H)$ and $Q$ a $N$-point subset of $H$, for any $0 < \epsilon < 1$, for each $k \in \mathbb{N}$ satisfying the so called Johnson-Lindenstrauss constraint*

$$k \geq \frac{24 \log N}{3\epsilon^2 - 2\epsilon^3},$$

*there exists a linear map $f^{\{Q,M,k\}} : H \to \mathbb{R}^k$ that embeds $Q$ into $\mathbb{R}^k$ in an almost isometric manner. More specifically, we have that*

$$(1 - \epsilon) \|\mathbf{a}_1 - \mathbf{a}_2\|_H^2 \leq \left\| f^{\{Q,M,k\}}(\mathbf{a}_1) - f^{\{Q,M,k\}}(\mathbf{a}_2) \right\|^2 \leq (1 + \epsilon) \|\mathbf{a}_1 - \mathbf{a}_2\|_H^2,$$

*for each $\mathbf{a}_1, \mathbf{a}_2 \in Q$.*

**Remark 1** *When there is no need to specify the dependence of $f^{\{Q,M,k\}}$ from $Q$, $M$, or $k$, we will omit them up to referring to is simply as $f$. Finally, we define $f^* : \mathbb{R}^k \to H$ to be the adjoint map of $f$ with respect to a fixed inner product $\langle \cdot, \cdot \rangle$ in $\mathbb{R}^k$.*

To apply such Lemma in our context, we select $M \geq 0$, equip $T^M(\mathbb{R}^d)$ with an inner product such that

$$\langle \mathbf{e}_{i_1} \otimes \cdots \otimes \mathbf{e}_{i_M}, \mathbf{e}_{j_1} \otimes \cdots \otimes \mathbf{e}_{j_M} \rangle := \delta_{i_1 j_1} \cdots \delta_{i_M j_M},$$

where $\{\mathbf{e}_{i_1} \otimes \cdots \otimes \mathbf{e}_{i_M}\}_{i_1,\dots,i_M \in \{1,\dots,d\}}$ is the canonical basis of $T^M(\mathbb{R}^d)$. Therefore, we have that $(T^M(\mathbb{R}^d), \langle \cdot, \cdot \rangle)$ is an Hilbert space.

**Theorem 5 (Existence and Uniqueness of the Signature, Lemma 2.10 (Terry J. Lyons, 2004))** *The following controlled differential equation*

$$d\mathbf{S}_t = \sum_{i=1}^{d} \mathbf{S}_t \otimes e_i dX_t^i, \quad \mathbf{S}_0 = \mathbf{1}$$

*has a unique solution which, at each $t \in [0, T]$ is the Signature $\mathbf{S}_t$ of $X$ on $[0, t]$.*

**Notation 1** *When there is no ambiguity about $X$, we will refer to its Signature as $\mathbf{S}$, while we use $\mathbf{S}^X$ if more paths are involved.*

**Theorem 6 (Theorem 2.29, (Terry J. Lyons, 2004))** *Given a continuous piecewise smooth path $X : [0, T] \to \mathbb{R}^d$, its Signature is the banal Signature $\mathbf{1} := (1, 0, 0, \cdots)$ if and only if $X$ is tree-like. In particular $\mathbf{S}^{X * \overleftarrow{X}} = 1$.*

Now, we provide results to show the relevance of these features. First of all, the following theorem ensures that the Signature of a path encodes its essence and characterizes it completely.

According to Theorem 6, the concatenation $X * \overleftarrow{X}$ of $X$ with its inverse[1] $\overleftarrow{X}$ has the same Signature as the constant path, but cannot be reparametrised to be constant. Similarly, if $X, Y, Z$ are non-constant paths, then $\mathbf{S}^{X*Y*\overleftarrow{Y}*Z*\overleftarrow{Z}*\overleftarrow{X}} = \mathbf{1}$, but $X * Y * \overleftarrow{Y} * Z * \overleftarrow{Z} * \overleftarrow{X}$ is not a path of the form $\gamma * \overleftarrow{\gamma}$ for any path $\gamma$. The formal definition of tree-like path is given in Definition 10.

**Theorem 7 (Characterizing Nature of the Signature, Corollary 1.4 (Ben Hambly, 2010))**
*Given a couple of continuous piecewise smooth paths $X$ and $\hat{X}$, then $\mathbf{S}^X = \mathbf{S}^{\hat{X}}$ if and only if $X * \overleftarrow{\hat{X}}$ is tree-like.*

This result is actually much stronger as it implies that the solution of any differential equation controlled by $X$ is fully determined by the vector fields and $\mathbf{S}$. In particular, Theorem 1 shows that the solution of any differential equation controlled by $X$ is essentially linear in $\mathbf{S}$.

Adapting Theorem III.8. (Cuchiero et al., 2021b) as done in (Cuchiero et al., 2021a) one obtains:

**Theorem 8 (Randomized Signature)** *For any fixed integer $M \geq 0$, any fixed partition $\mathcal{D} = \{t_0, \cdots, t_N\}$ of $t$ such that $0 \leq t_0 < \cdots < t_N \leq T$, let us consider the Truncated Signature of order $M$ of $X$ at such times, that is $Q := \{\mathbf{S}_{t_0}^M, \cdots, \mathbf{S}_{t_N}^M\}$ such that its elements all lie in $\left(T^M\left(\mathbb{R}^d\right), \langle \cdot, \cdot \rangle\right)$. Let us now select $0 < \epsilon < 1$, $k \in \mathbb{N}$ satisfying the associated Johnson-Lindenstrauss constraint, let $f$ be the implied Johnson-Lindenstrauss map and $f^*$ its adjoint map. Then, the solution of the controlled differential equation in $\mathbb{R}^k$*

$$dZ_t = \sum_{i=1}^d f\left(f^*\left(Z_t\right)e_i\right) dX_t^i, \quad Z_0 = f(\mathbf{1})$$

*on $\mathcal{D}$, that is $\{Z_{t_0}, \cdots, Z_{t_N}\}$, are called the Randomized Signature of $X$ on $\mathcal{D}$. It holds that each $Z_{t_k}$ is the projection of $\mathbf{S}_{t_k}^M$ from $\left(T^M\left(\mathbb{R}^d\right), \langle \cdot, \cdot \rangle\right)$ onto $\mathbb{R}^k$ via $f$. To conclude, since $f^*\left(Z_{t_k}\right)$ is close to $\mathbf{S}_{t_k}^M$ in the norm of $T^M\left(\mathbb{R}^d\right)$, we conclude that $Z$ preserves the geometric properties and the approximation power of $S$.*

Adapting Theorem III.7 in (Cuchiero et al., 2021b) it can be be shown that (asymptotically) the JL projected vector fields stem from random matrices:

**Theorem 9 ($Z$ is a random dynamical system)** *For $k \longmapsto \infty$, for each $i \in \{1, \cdots, d\}$ the linear vector fields $f^{\{k\}}\left(f^{\{k\}*}(\cdot)e_i\right) : \mathbb{R}^k \to \mathbb{R}^k$ are square matrices with asymptotically normally distributed, independent entries.*

Leveraging this results, we define the following:

**Definition 10 (Localized Randomized Signature)** *For any random matrices $A_1, \ldots, A_d$ in $\mathbb{R}^{k \times k}$ and shifts $b_1, \ldots, b_d$ in $\mathbb{R}^{k \times 1}$ such that maximal non-integrability holds on a random starting point $z \in \mathbb{R}^k$, any fixed activation function $\sigma$, and $d$-dimensional control $X$, the solution of*

$$dZ_t = \sum_{i=1}^d \sigma\left(A_i Z_t + b_i\right) dX_t^i, \quad Z_0 = z, \quad t \in [0, T]. \tag{7}$$

*is called the Localized Randomized Signature of $X$ and we often refer to it as Randomized Signature.*

**Theorem 10 (Signature is a Reservoir (Restatement of Theorem 1))** *Let $V_i : \mathbb{R}^m \to \mathbb{R}^m, i = 1, \ldots, d$ be vector fields regular enough such that $dY_t = \sum_{i=1}^d V^i\left(Y_t\right) dX_t^i, Y_0 = y \in \mathbb{R}^m$, admits a unique solution $Y_t : [0, T] \to \mathbb{R}^m$. Then, for any smooth test function $F : \mathbb{R}^m \to \mathbb{R}$ and for every $M \geq 0$ there is a time-homogeneous linear operator $L : \mathcal{T}^M\left(\mathbb{R}^d\right) \to \mathbb{R}$ which depends only on $(V_1, \ldots, V_d, F, M, y)$ such that $F\left(Y_t\right) = L\left(\mathbf{S}_t^M\right) + \mathcal{O}\left(t^{M+1}\right)$, and $t \in [0, T]$.*

---

[1] $\overleftarrow{X}_t := X_{T-t}$

**Proof:**   *We provide just a sketch. For every smooth test function $F : \mathbb{R}^m \to \mathbb{R}$, the formal Taylor expansion along controls together with the fundamental theorem of calculus gives us that*

$$F\left(Y_t\right) = F(y) + \sum_{i=0}^{d} \int_0^t V_i F\left(Y_s\right) dX_s^i, \quad t \geq 0. \tag{8}$$

*For $y \in \mathbb{R}^m$, this equation can be inserted into itself leading to a generalized version of Taylor expansion for controlled ordinary differential equations*

$$\sum_{k=0}^{M} \sum_{(i_1,\ldots,i_k)\in\{0,\ldots,d\}^k} V_{i_1} \ldots V_{i_k} F(y) \int_{0 \leq t_1 \leq \ldots \leq t_k \leq t} dX_{t_1}^{i_1} \ldots dX_{t_k}^{i_k} + R_M(F,t)$$

*for $M \geq 0$, with the remainder satisfying*

$$R_M(F,t) = \sum_{(i_1,\ldots,i_{M+1})\in\{0,\ldots,d\}^{M+1}} \int_{0 \leq t_1 \leq \ldots \leq t_{M+1} \leq t} V_{i_1} \ldots V_{i_{M+1}} F\left(Y_{t_1}\right) dX_{t_1}^{i_1} \ldots dX_{t_{M+1}}^{i_{M+1}}.$$

$\square$

### A.3   PRACTICAL EXAMPLE OF THE COMPUTATION OF SIGNATURES (ONE DIMENSION).

To enhance intuition, we exemplify the computation of Signatures in the simplest setting possible. Let $X : [0,T] \to \mathbb{R}$, then $S_t^1 = \int_0^t dX_s$ — which is exactly $X_t - X_0$. To get $S_t^2$, we instead have to compute the following iterated integral: $S_t^2 = \int_0^t \left(\int_0^v dX_s\right) dX_v$. Signatures of higher order $S_t^j$ are computed in a similar way, by iteratively integrating the path $j$ times. As a practical example, let $X_t = t$. Then it is easy to see that $S_t^j = \frac{t^j}{j!}$. Now let $Y(t)$ be an analytic function of time for which we have $Y(t) = \sum_{j=0}^{\infty} Y_0^{(j)} \frac{t^j}{j!}$. Taylor's theorem combined with the previous computation of the Signatures implies that $Y$ can be approximated as a linear map of the Signatures of $t$. Finally, note that $S_t^j$ in this case gets smaller and smaller in magnitude as $j$ increases. This suggests that the truncated Signature can be safely used to approximate $Y$.

**Computational Complexity and dimensionality of Signatures.**   The computational complexity for computing $\mathbf{S}_t^M$ is $\mathcal{O}(d^M)$. Indeed, consider $d = 2$: $S_t^2$ is a $2 \times 2$ matrix with elements $\int_0^t \left(\int_0^v dX_s^1\right) dX_v^1$, $\int_0^t \left(\int_0^v dX_s^1\right) dX_v^2$, $\int_0^t \left(\int_0^v dX_s^2\right) dX_v^1$ and $\int_0^t \left(\int_0^v dX_s^2\right) dX_v^2$. For $M = 3$, the object to compute is instead a $2 \times 2 \times 2$ tensor, containing all integrals of the type $\int_0^t \left(\int_0^w \left(\int_0^v dX_s^{i_1}\right) dX_v^{i_2}\right) dX_w^{i_3}$ for all $i_1, i_2, i_3 \in \{1,2\}$. Hence, the complexity — as well as the features dimensionality — scales exponentially in $M$.

# B  EXPERIMENTS

In this section, we prove that our approach can accurately learn the dynamics of several rough differential equations. First of all, we focus on dynamics of 1-dimensional stochastic Double Well which is characterized by a highly nonlinear dynamics. Secondly, we study the multidimensional case of the 4-dimensional Ornstein–Uhlenbeck. Then, we investigate the performance on the rough dynamics of Fractional Geometric Brownian Motion even for choices of extremely low values of the hurst parameter $H$. We also obtain promising results in learning the dynamics of an Enzyme-Substrate Reactions. Next, we show how in high dimensions randomized signatures are computationally efficient, and also present one experiment in comparison with neural controlled differential equations (Kidger et al., 2020). To conclude, we show how the randomized signature approach can be used in cases where the time grid is irregularly sampled.

## B.1  PRELIMINARY RESULTS: ROUGH DIFFERENTIAL EQUATIONS

**1-Dimensional Stochastic Double Well.**  Let us recall that the dynamics of the 1-Dimensional Stochastic Double Well process is given by

$$dY_t = \theta Y_t \left( \mu - Y_t^2 \right) dt + \sigma dW_t, \quad Y_0 = y_0 \in \mathbb{R}, \quad t \in [0, 1].$$

where $W_t$ is a 1-dimensional Brownian motion, and $(\mu, \theta, \sigma) \in \mathbb{R} \times \mathbb{R}^+ \times \mathbb{R}^+$. Let us fix $y_0 = 1$ and $(\mu = 2, \theta = 1, \sigma = 1)$, and the partition $\mathcal{D}$ of $[0, 1]$ to have $N = 101$ equally spaced time steps. We train a Ridge Regression with regularization parameter $\lambda = 0.001$ to map instances of Randomized Signature of the controls $Z$ into the respective solution $Y_t$. We repeat the experiment on different values of the number $N_{\text{Train}}$ of train samples and dimension $k$ of the Randomized Signature. On the left of Figure 9, we plot an example of the trajectory of $Z$ while, on its right, we plot the comparison of the true and the generated time series on an out-of-sample case. The following table shows the performance in terms of $L^2$ relative error on 10000 test samples:

| | $N_{\text{Train}} = 1$ | $N_{\text{Train}} = 10$ | $N_{\text{Train}} = 100$ | $N_{\text{Train}} = 1000$ | $N_{\text{Train}} = 10000$ |
|---|---|---|---|---|---|
| $k = 111$ | **0.319443** | **0.100310** | **0.007727** | 0.005854 | 0.005932 |
| $k = 222$ | 0.447563 | 0.357439 | 0.030950 | **0.005193** | 0.004558 |
| $k = 444$ | 0.504822 | 0.46000 | 0.092313 | 0.005690 | **0.0043893** |

Table 1: Double Well: Relative $L^2$ Error

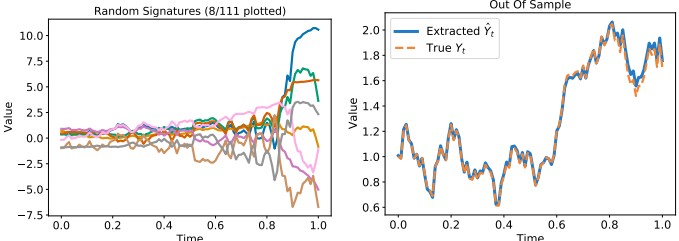

Figure 9: Double Well: Randomized Signatures (left) - Test Sample (right).

**4-Dimensional Ornstein–Uhlenbeck process.**  Let us recall that the dynamics of the 4-Dimensional Ornstein–Uhlenbeck process is given by

$$d\mathbf{Y}_t = (\boldsymbol{\mu} - \boldsymbol{\Theta} \mathbf{Y}_t) dt + \boldsymbol{\Sigma} d\mathbf{W}_t, \quad \mathbf{Y}_0 = \mathbf{y}_0 \in \mathbb{R}^4, \quad t \in [0, 1]$$

where $\mathbf{W}_t$ is a 4-dimensional Brownian motion, and $(\boldsymbol{\mu}, \boldsymbol{\Theta}, \boldsymbol{\Sigma}) \in \mathbb{R}^4 \times \mathbb{R}^{4 \times 4} \times \mathbb{R}^{4 \times 4}$. Let us fix $\mathbf{y}_0 = \mathbf{1}, \boldsymbol{\mu} = \mathbf{1}, \boldsymbol{\Sigma} = \mathbb{1}_4, \boldsymbol{\Theta}_{i \cdot j} = i/j$, the partition $\mathcal{D}$ of $[0, 1]$ to have $N = 101$ equally spaced time steps, and $k = 708$. Finally, we train a Ridge Regression with $\lambda = 0.001$ on $N_{\text{Train}}$ train sample and Figure 10 shows the comparison of out-of-sample generated and true trajectories while Table 2 reports the performance in terms of $L^2$ relative error on 10000 test samples.

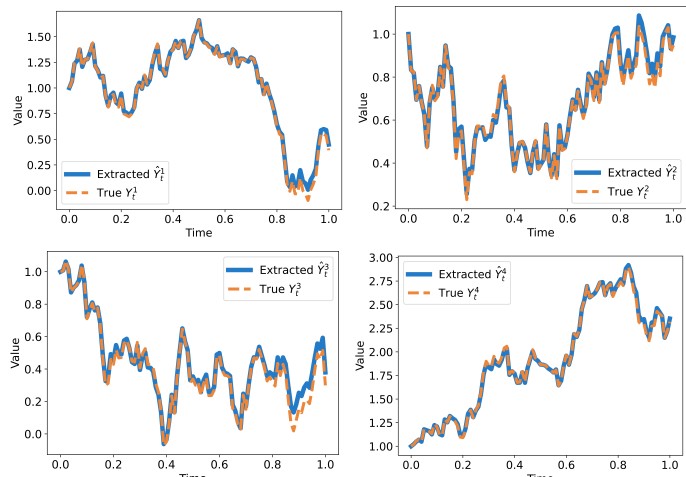

Figure 10: 4-Dimensional Ornstein–Uhlenbeck process: Test Sample.

| $k = 708$ | $N_{\text{Train}} = 1$ | $N_{\text{Train}} = 10$ | $N_{\text{Train}} = 100$ | $N_{\text{Train}} = 1000$ | $N_{\text{Train}} = 10000$ |
|---|---|---|---|---|---|
| $Y^1$ | 0.025960 | 0.010615 | 0.005145 | 0.001467 | 0.000929 |
| $Y^2$ | 0.047323 | 0.023858 | 0.009432 | 0.001838 | 0.001247 |
| $Y^3$ | 0.033357 | 0.024392 | 0.009694 | 0.002008 | 0.001228 |
| $Y^4$ | 0.027637 | 0.020390 | 0.007798 | 0.001775 | 0.001140 |

Table 2: 4-Dimensional Ornstein–Uhlenbeck process: Relative $L^2$ Error.

**1-Dimensional Geometric Fractional Brownian Motion.**  Let us recall that the dynamics of the 1-Dimensional Geometric Fractional Brownian Motion process given by

$$dY_t = Y_t \left( \mu dt + \sigma dB_t^{(H)} \right), \quad Y_0 = y_0 \in \mathbb{R}^+,$$

where $\mu \in \mathbb{R}$, $\sigma \in \mathbb{R} \setminus \{0\}$, and $B_t^{(H)}$ is a one-dimensional fractional Brownian motion of hurst parameter $H \in (0, 1)$. In this experiment, we fix $y_0 = 1$ and $(\mu = 1, \sigma = 1)$, we let $H$ vary in $\{0.1, 0.2, 0.3, 0.4, 0.5\}$, and the partition $\mathcal{D}$ of $[0, 1]$ is made of $N$ equally spaced times. We highlight that, as per Theorem D.3.2 in (Biagini et al., 2008), when $H < 1/4$, the fractional Brownian motion is so rough that it is no longer a Rough Path and its Signature is no longer defined. Finally, we train a Ridge Regression with $\lambda = 0.001$ on $N_{\text{Train}} = 10000$ train samples of $Z$ into the logarithm of the respective solution $\log(Y_t)$ which implies that we need to then apply the $\exp$ function to our predictions. The right of Figure 11 shows the comparison of the generated and true trajectories on an out-of-sample trajectory while the following table shows the Relative $L^2$ Error as we vary the number of time steps $N$ and hurst parameter $H$:

| | $H = 0.1$ | $H = 0.2$ | $H = 0.3$ | $H = 0.4$ | $H = 0.5$ |
|---|---|---|---|---|---|
| $N = 11$ | $2.44 \cdot 10^{-3}$ | $7.07 \cdot 10^{-4}$ | $5.73 \cdot 10^{-5}$ | $5.79 \cdot 10^{-6}$ | $2.72 \cdot 10^{-8}$ |
| $N = 101$ | $3.04 \cdot 10^{-4}$ | $6.96 \cdot 10^{-5}$ | $1.58 \cdot 10^{-5}$ | $1.76 \cdot 10^{-6}$ | $3.93 \cdot 10^{-8}$ |
| $N = 1001$ | $3.78 \cdot 10^{-5}$ | $1.11 \cdot 10^{-5}$ | $4.03 \cdot 10^{-6}$ | $1.46 \cdot 10^{-6}$ | $1.32 \cdot 10^{-6}$ |

Table 3: Relative $L^2$ Error Fractional Geometric Brownian Motion.

**Enzyme-Substrate Reactions.**  First, we consider a set of highly nonlinear CODEs modeling the interaction between a catalyst (enzyme, with abundance $C_t$) and a substrate S (with abundance $S_t$). Substrate (e.g. lactose) is injected via the control variable $X_t$, reacts with the enzyme (e.g. lactase)

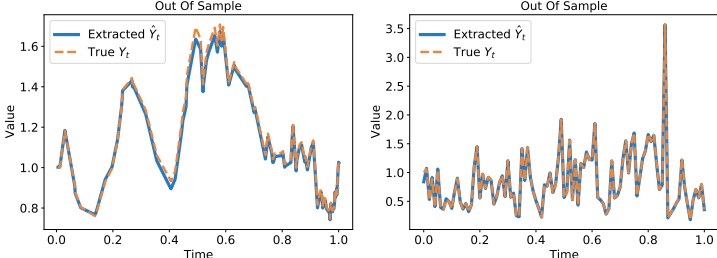

Figure 11: Double Well: Irregularly Sampled Test Sample (left) - Fractional Geometric Brownian Motion Test Sample (right).

and the concentration $Y_t$ of the product chemical (e.g. glucose) is observed (Ingalls, 2013).

$$\begin{cases} dS_t & = (C_t - k_1 S_t (1 - C_t)) \, dt + X_t dt \\ dC_t & = - (C_t - k_1 S_t (1 - C_t)) \, dt - k_2 C_t dt \\ dY_t & = k_2 C_t dt. \end{cases}$$

Following (Ingalls, 2013) we choose $(k_1, k_2) = (30, 10)$, set $(S_0, P_0, Y_0) = (0, 0, 0)$ and consider the evolution on $0 \le t \le 1$. We fix the time grid to have $N = 101$ equally spaced time steps and the control $X_t$ to follow the law of $W_t^2$ where $W_t$ is a 1-dimensional Brownian Motion (to ensure positivity). We train a Ridge Regression with regularization parameter $\lambda = 10^{-3}$ to map $10^5$ instances of 222-dimensional Randomized Signature $Z$ of the controls into the respective solution $Y_t$. On the left of Figure 12, we plot the comparison of the true and the generated time series on a test sample. As we can see, the model has learnt to correctly map a trajectory of $X_t$ to the respective system response $Y_t$. More surprisingly, the right of such a figure shows that our model is able to predict the correct output even if we stimulate the system with substrate injection that follows a completely different law such as $0.5 \cdot \mathbb{1}_{\{W_t^2 > 0.5\}}$. This suggests that the system was correctly identified out of distribution.

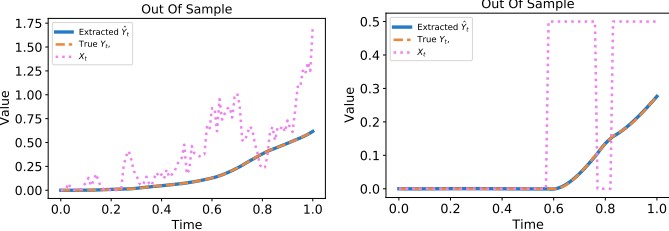

Figure 12: Enzyme-Substrate Reactions stimulated with Squared Brownian Motion (left) - Step Function (right).

**Comparison with Neural Controlled Differential Equations** In this experiment, we benchmark the performance of our Randomized Signature model with that of Neural Controlled Differential Equations (NCDEs) (Kidger et al., 2020), a recently proposed competitive deep learning technique for controlled dynamical system modelling. We test the two models on the task of learning the dynamics of the 1-dimensional Ornstein-Uhlenbeck Process. As usual, to train our model, we fix an equally spaced partition $\mathcal{D}$ of $[0, 1]$, extract the Randomized Signature of the controls for $N_{Train}$ samples, and train a Ridge Regression to map them into the respective integrated solutions. The NCDE models instead parametrize the potentials of a latent controlled differential equation with feed-forward neural networks with 1 hidden layer of $n_{nodes}$. Figure 13 shows the comparison of our model against several version of the NCDEs. Additionally, Table 4 and Table 5 show the number of trainable parameters, average $L^2$ Relative Errors, and training times for all the models for $N_{Train} = 100$ and $N_{Train} = 1000$ respectively. We observe that our model is the best one from all aspects and that not even a considerable amount of training data is beneficial for NCDEs.

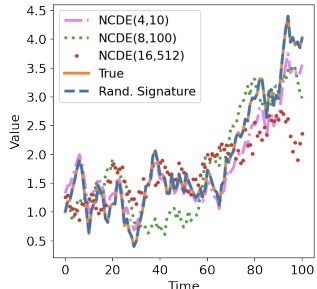

Figure 13: Out-of-Sample comparison of NCDE($n_{channels}, n_{nodes}$) models, Randomized Signature model and Ground Truth.

|  | Rand. Sig Model | NCDE(4,10) | NCDE(8,100) | NCDE(16,512) |
|---|---|---|---|---|
| Trainable Param | **107** | 107 | 1733 | 16961 |
| Training Time | **0.047390** | 1833.00 | 1837.858 | 2485.80 |
| Error | $\mathbf{1.23 \cdot 10^{-6}}$ | 0.081516 | 0.080829 | 0.0800839 |

Table 4: Randomized Signature vs. NCDEs - $d = 80$, $N_{Train} = 100$

|  | Rand. Sig Model | NCDE(4,10) | NCDE(8,100) | NCDE(16,512) |
|---|---|---|---|---|
| Trainable Param | **107** | 107 | 1733 | 16961 |
| Training Time | **0.238423** | 2853.00 | 3972.79 | 6510.95 |
| Error | $\mathbf{2.45 \cdot 10^{-7}}$ | 0.076195 | 0.072763 | 0.073845 |

Table 5: Randomized Signature vs. NCDEs - $d = 80$, $N_{Train} = 1000$

## B.2 ADDITIONAL DETAILS

**Learning the Signatures.** In this experiment, we fix the partition $\mathcal{D} = \{t_0, \cdots, t_N\}$ of $[0, 1]$ to have 201 equally-spaced time steps and the regularization parameter of the Ridge Regression is $\lambda = 0.001$.

**Reconstruction and Randomized Signature Autoencoders.** In this experiment, we fix the partition $\mathcal{D} = \{t_0, \cdots, t_N\}$ of $[0, 1]$ to have 101 equally-spaced time steps, fix $k = 111$, and consider a single realization $X_{Train}$ of an Ornstein–Uhlenbeck process of asymptotic mean $\mu = 2$, mean reversion $\theta = 1$, and volatility $\sigma = 1$. We train a Ridge Regression with regularization parameter $\lambda = 0.001$ to map the Randomized Signature $Z$ of the one and only train sample $X_{Train}$ into $X_{Train}$ itself.

|  | $k = 111$ | $k = 222$ | $k = 444$ |
|---|---|---|---|
| $N_{Test} = 10000$ | $8.35 \cdot 10^{-9}$ | $2.26 \cdot 10^{-8}$ | $2.27 \cdot 10^{-9}$ |

Table 6: Relative $L^2$ Error Autoencoder.

**Comparison with Vanilla Signature Approach** Let us recall that the dynamics of the $d$-Dimensional Fractional Ornstein–Uhlenbeck process is given by

$$d\mathbf{Y}_t = (\boldsymbol{\mu} - \boldsymbol{\Theta}\mathbf{Y}_t)\, dt + \boldsymbol{\Sigma}d\mathbf{B}_t^H, \quad \mathbf{Y}_0 = \mathbf{y}_0 \in \mathbb{R}^d, \quad t \in [0, 1]$$

where $\mathbf{B}_t^H$ is a $d$-dimensional Fractional Brownian motion of hurst parameter $H \in (0, 1)$, and $(\boldsymbol{\mu}, \boldsymbol{\Theta}, \boldsymbol{\Sigma}) \in \mathbb{R}^d \times \mathbb{R}^{d \times d} \times \mathbb{R}^{d \times d}$. Let us fix $\mathbf{y}_0 = \mathbf{1}$, $\boldsymbol{\mu} = \mathbf{1}$, $\boldsymbol{\Sigma} = \mathbb{1}_d$, $\boldsymbol{\Theta}_{i.j} = i/j$, the partition $\mathcal{D}$ of $[0, 1]$ to have $N = 101$ equally spaced time steps, and $H = 0.3$.

**1-Dimensional Stochastic Double Well - Irregularly Sampled Time Grid.** Let us recall that the dynamics of the 1-Dimensional Stochastic Double Well process is given by

$$dY_t = \theta Y_t \left( \mu - Y_t^2 \right) dt + \sigma dW_t, \quad Y_0 = y_0 \in \mathbb{R}, \quad t \in [0, 1] \tag{9}$$

where $W_t$ is a 1-dimensional Brownian motion, and $(\mu, \theta, \sigma) \in \mathbb{R} \times \mathbb{R}^+ \times \mathbb{R}^+$. For each train and test sample, the partition $\mathcal{D}$ of $[0, 1]$ is made of $N$ randomly drawn times. More precisely, $\mathcal{D} = \{0, t_1, \cdots, t_{N-1}, 1\}$ such that $t_k = 1/(1 - \exp(-s_k))$ and $\{s_1, \cdots, s_{N-1}\}$ are $N - 2$ independent realizations of a uniform distribution $\mathcal{U}[0, 1]$ sorted increasingly. As a result, the probability that two samples share the same $\mathcal{D}$ is null. We train a Ridge Regressor with $N_{\text{Train}} = 10000$ train samples and Figure 11 shows the comparison of an out-of-sample generated and true trajectory. Finally, Table 7 shows the Relative $L^2$ Error on 10000 test samples as we vary the number of time steps $N$ and $k$ and while comparing it to the respective experiment in case the time grid is regularly spaced. As we can see, even though the performance are worse than the regularly sampled setup, this technique proves to be anyway reliable on irregularly sampled regimes. In this experiment, we fix $y_0 = 1$ and $(\mu = 2, \theta = 1, \sigma = 1)$ while the regularization parameter of the Ridge Regression is $\lambda = 0.001$.

|  | $(N, k) = (11, 111)$ | $(N, k) = (101, 222)$ | $(N, k) = (1001, 332)$ |
|---|---|---|---|
| Irregular | 0.082735 | 0.016885 | 0.010902 |
| Regular | 0.026759 | 0.004465 | 0.003004 |

Table 7: Irregularly Sampled Double Well: Relative $L^2$ Error.

**Tumor Growth Kinetics.** We set $(k_1, k_2, \lambda_0, \lambda_1, \Psi) = (10, 0.5, 0.9, 0.7, 20)$, and $\left( u_0^1, u_0^2, u_0^3, u_0^4, Y_0 \right) = (2, 0, 0, 0, 2)$. We fix the partition $\mathcal{D}$ of $[0, 10]$ to have 101 equally spaced time steps and we again train a Ridge Regression with regularization parameter $\lambda = 0.001$ to map 10000 instances of the 222-dimensional Randomized Signature $Z$ of the controls into the respective solution $Y_t$.

**Electrochemical Battery Model.** In this experiment, we fix the partition $\mathcal{D}$ of $[0, 500]$ to have 1001 equally spaced time steps, the regularization parameter of the Ridge Regression is $\lambda = 10^{-3}$ while $k \in \{50, 166, 1000\}$. Regarding the latest experiment, we simply learn to map the Random Signature of the controls into the respective $Y$ to which we added white noise with variance 0.01. Regarding NNARX, for the regressor we follow the results presented in (Masti and Bemporad, 2018), where $n_a = n_b = 12$, and $n_k = 1$ lead to the best results. Regarding the neural network, we use a feedforward neural network with input dimension $n_a + n_b + 1 = 25$ and 2 hidden layers each with either 22 (NNARX22) or 1000 (NNARX1000) hidden units. For the loss function, we used the mean square error optimized using Adam with learning rate 0.01 for 100 epochs. Table 8 reports the performances in terms of MSE on 1000 test samples for our models compared to NNARX.

|  | $k = 50$ | $k = 166$ | $k = 1000$ | NNARX 22 | NNARX 1000 |
|---|---|---|---|---|---|
| $N_{Test} = 1000$ | $3.15 \cdot 10^{-4}$ | $2.66 \cdot 10^{-4}$ | $\mathbf{2.59 \cdot 10^{-4}}$ | $3.88 \cdot 10^{-4}$ | $2.62 \cdot 10^{-4}$ |

Table 8: Electrochemical Battery Model: MSE Error Comparison.

**Time Series Classification**

DATASET DETAILS We run our TSC experiments on the UCR archive, one of the most popular benchmarks for TSC methods (Dau et al., 2019; Fawaz et al., 2019). It consists of 128 heterogeneous datasets of different sizes comprising possibly irregularly sampled one-dimensional time series of various lengths. Rocket is one of the best performing algorithms on this benchmark.

ROCKET When implementing Rocket, we generate $k$ kernels each characterized by a certain random length $l_{filter}$, random weights $\omega \in \mathbb{R}^{l_{filter}}$, and random jump j. Since the lengths and the jumps are randomly sampled, the length $M$ of the vector $\mathbf{Z} = (Z^1, \cdots, Z^M)$ of the random features varies accordingly. Given that the number of possible combinations of the values of $l_{filter}$ and

j is limited, we can group the kernels based on such combinations. As a result, among each of these groups, the length $M$ is fixed and, additionally, each kernel is applied to the very same sub-time-series $\left(X_i, X_{i+\mathrm{j}}, \cdots, X_{i+(l_{filter}-1)\times\mathrm{j}}\right)$ of $X$ starting from position $i$. If we now focus on one of these subgroups and let us assume it contains $\tilde{k}$ kernels of length $l_{filter}$ and jump j. As a result of the application of each kernel on $X$, we obtain the following matrix

$$\begin{bmatrix} Z^{0,1} & Z^{1,1} & \dots & \dots & Z^{M,1} \\ Z^{0,2} & Z^{1,2} & \dots & \dots & Z^{M,2} \\ \vdots & \vdots & \vdots & \ddots & \vdots \\ Z^{0,\tilde{k}} & Z^{1,\tilde{k}} & \dots & \dots & Z^{M,\tilde{k}} \end{bmatrix}$$

where $Z^{i,r}$ is the result of the action of the $r$-th kernel on the sub-time-series $\left(X_i, X_{i+\mathrm{j}}, \cdots, X_{i+(l_{filter}-1)\times\mathrm{j}}\right)$ of $X$ starting from position $i$. Finally, out of each row, we extract the features described above.

Turning to our approach, we generate $k$ kernels each characterized by a certain length $l_{filter}$, weights $\omega = (\omega_0, \omega_1) \in \mathbb{R}^2$, and jump j. As before, let us again group them based on all the possible combinations of $l_{filter}$ and j and focus on only one of such subgroups. As a consequence, we have $\tilde{k}$ kernels each characterized by weights $(\omega_0^r, \omega_1^r)$, $r \in \{1, \cdots, \tilde{k}\}$ with fixed length and jump. The application of each of these $\tilde{k}$ kernels following the update rules proposed in Algorithm 1 is equivalent to extracting a 1-dimensional Randomized Feature of each of the sub-time-series $\left(X_i, X_{i+\mathrm{j}}, \cdots, X_{i+(l_{filter}-1)\times\mathrm{j}}\right)$. As a result, we obtain the following matrix

$$\begin{bmatrix} Z^{0,1} & Z^{1,1} & \dots & \dots & Z^{M,1} \\ Z^{0,2} & Z^{1,2} & \dots & \dots & Z^{M,2} \\ \vdots & \vdots & \vdots & \ddots & \vdots \\ Z^{0,\tilde{k}} & Z^{1,\tilde{k}} & \dots & \dots & Z^{M,\tilde{k}} \end{bmatrix}$$

where $Z^{i,r}$ is the 1-dimensional Randomized Signature of $\left(X_0, X_{\mathrm{j}}, \cdots, X_{(l_{filter}-1)\times\mathrm{j}}\right)$ via the $r$-th kernel. Let us we define the random matrix $A$ guiding the evolution of the Randomized Signature as the $\tilde{k} \times \tilde{k}$ diagonal matrix having the weights $\omega_0^r$ on the diagonal while the random bias $b$ is a column vector having the $\omega_1^r$ as entries. Then, the $i$-th column is the $\tilde{k}$-dimensional Randomized Signature of the sub-time-series $\left(X_i, X_{i+\mathrm{j}}, \cdots, X_{i+(l_{filter}-1)\times\mathrm{j}}\right)$. Finally, out of each row, we extract the features described above.

**Comparison of Computational Complexities and Dimensions**   The following table reports key figures regarding the computational complexity and dimensionality of both Signature and Randomized Signature.

|  | Comp. complexity | Dimension | Guarantees |
|---|---|---|---|
| **Truncated Sign. ($M$- components):** | $\mathcal{O}(d^M)$ | $\mathcal{O}(d^M)$ | ✓(Thm 1) |
| **Randomized Sign. ($d$ - dimensional):** | $\mathcal{O}(k^2 d)$ | $O(k)$ | ✓(Thm 2) |

Table 9: Properties of (Randomized) Signatures of a $d$-dimensional path.

