# OpenReview forum: "Randomized Signature Layers for Signal Extraction in Time Series Data"
_ICLR.cc/2022/Conference — ICLR 2022 Submitted_

### Official Review · Reviewer_g484 · 2021-10-26

**Correctness:** 3
**Technical Novelty And Significance:** 2
**Empirical Novelty And Significance:** 3
**Recommendation:** 5
**Confidence:** 4

**Main Review:**

Overall this paper is very well written and tackles an interesting problem - dealing with the blowup in the dimensionality of the signature.

A lot of recent papers in the ML literature that utilise the signature pay little notice of the severe feature blowup associated with increasing signature feature dimension. The result of this is a number of methods that do not scale well to tasks with large numbers of feature channels. This paper outlines an interesting method of dealing with this by way of random projections applied to the path

## Strengths
- Well written with a good introductory section to signatures and their usefulness.
- Randomised signature layers are an interesting way to reduce signature dimension - a current bottleneck for a lot of signature methodologies.

## Weaknesses
- **Method is not entirely novel** - Lyons et al. "A feature set for streams and an application to high-frequency financial tick data" introduce this idea of randomised signatures and it is considered in Morrill et al. "A Generalised Signature Method for Multivariate Time Series Feature Extraction". Neither of these papers are mentioned/referenced by the authors.
- **Performance on time series classification tasks seems poor** -  Whilst the "rocket_ours" classifier comes second of all classifiers considered, "rocket" was already the top performer. I'm not sure how I should be considering this other than that the randomised signature features make the rocket classifier worse -- why would I use randomised signature features then?
- **Informal theorem is too informal** - The randomised signature theorem with the claim "has comparable approximation power as the Signature" needs more explanation. Whilst I am a fan of informal theorem statements in such papers in general, I need some indication of what such a statement actually means. This can be easily remedied by a high level explanation of what is going on, or a mathematical statement as to what this comparable approximation power actually means. It also needs to reference a full, formal theorem somewhere.

**Summary Of The Paper:**

The authors utilise random projections before computation of the signature resulting in "Randomised signatures" from which similar information can be extracted (compared with the standardly applied signature transform) with reduced computational burden and lower overall feature dimension.

The authors demonstrate the utility of this randomised methodology on a variety of tasks including learning rough differential equations and standard time series classification benchmarks.

**Summary Of The Review:**

This paper would represent the best reference to the idea of "randomised signature" features if accepted. However, it is currently lacking in that it does not adequately reference prior work, and it has theorems that are slightly too vague. If these things can be rectified, I would consider raising my score.

---

> ### Author Response · Authors · 2021-11-18
> **Answer to your concerns**
>
> Thanks for the nice comments on our paper. We provide an answer to your questions below
>
> &nbsp;
>
> - ***Randomized signatures were already proposed in Lyons et al. 2020 and Morril et al. 2021***
>
> The first work you mention does not discuss Randomized Signatures or random projections (the keyword "random" is not even to be found in the pdf). The second paper you mention discusses random projections of the controls into a lower-dimensional space, to then extract the truncated signature. Our work is fundamentally different: we keep the control as it is and extract a random projection of its Signature. We added a couple of lines in the related work, but kindly ask the reviewer to update their comment.
>
> &nbsp;
>
> - ***Performance on time series classification is not state-of-the-art***
>
> We never mention that our goal is to deliver state-of-the-art performance on classification. Indeed, this section only takes a single page. As stated in the introduction, the purpose of our work is to showcase the expressiveness of Randomized Signatures as general-purpose features for machine learning tasks, with strong theoretical grounding. Not only Rocket (the only method that gives better performance for classification) lacks a rigorous theory, but also cannot be straightforwardly used for tasks such as black-box system identification. While further research and engineering are needed to outperform Rocket (the version we implemented does not include any additional “tricks”), we claim our Randomized Signature-based classifier is exceptionally interesting: its vanilla version already outperforms deep learning techniques, and the underlying features can be used for arbitrary tasks.
>
> &nbsp;
>
> - ***Formal discussion of the theorem***
>
> Thanks for the suggestion. We added a few lines after the theorem to enhance intuition and clarify the theoretical statements. For a more thorough mathematical treatment of the subject, we refer the interested reader to the relevant work of Cuchiero 2021(a,b).

---

### Official Review · Reviewer_KwmV · 2021-11-01

**Correctness:** 3
**Technical Novelty And Significance:** 3
**Empirical Novelty And Significance:** 2
**Recommendation:** 5
**Confidence:** 4

**Main Review:**

Strengths:
S1: Following Theorem 1 theoretically justifies that Signature of $X$ is suited to solving differential equations driven by $X$.

S2: Following Theorem 2, Randomized Signature approximates Signature while being much cheaper to compute. The approximation power of Randomized Signature to the original Signature is empirically verified in Section 4.1 (in particular Figure 3).

S3: In Section 4.3, Randomized Signature can find solutions of differential equations that are close to ground-truth solutions.

S4: For time series classification, Randomized Signature when combined with Rocket (Dempster et al., 2020) gives better performance than existing related deep learning methods. It however does not perform as well as Rocket.

Weaknesses:
W1: The paper provides informal versions of Theorems 1 and 2 only. I cannot find the formal versions as well as their proofs in the appendix. I would recommend the authors to provide more details. This will help to make points such as "comparable approximation power as the Signature" in Theorem 2 more rigorous.

W2: How to choose activation $\sigma$? There seems also no explanation about this in Section 4.

W3: From time complexity analysis in Section 3.2, the runtime speed up seems significant. However, the experiment in Section 4.1 reveals using Randomized Signature yields less than an order of magnitude speed up (3 times in particular). This can be because $k$ is quite large compared to $M$. The authors should provide more detailed runtime comparisons between the two types of signature in the experiments in Section 4.3.

W4: Most experiments in Section 4.3 are on data with $d \leq 5$. How about data with higher number of dimensions such as more than 10? It would be interesting to see scalability to $d$ in both runtime and quality of Randomized Signature. Could the authors provide a summary of the 128 UCI data sets for time series classification? This will help to better understand impact of $d$ on the performance.

W5: Randomized Signature + Rocket performs worse than Rocket, due to limitations of the update scheme in Algorithm 1. The authors already provided some justifications in the last paragraph of Section 4.4.

**Summary Of The Paper:**

This paper proposes a Signature-based method for learning feature representation of time series, denoted as $d$-dimensional path $X$. By definition, Signature is a sequence of $M$ tensors where each tensor is a sum of integral terms defined on dimensions of  $X$. Theoretically Signature is guaranteed to linearly explain the solution of any differential equation driven by $X$ which implies that it encodes well $X$. Computing Signature however has time complexity $O(d^M)$ rendering it impractical for learning purposes. To alleviate the issue, the paper introduces Randomized Signature, which can be computed in $O(k^2 d)$ only where $k$ is the number of Randomized Signatures. Each of such Signature has $O(kd)$ dimensions. As stated by Theorem 2 in the paper, Randomized Signature approximates well the original Signature. Experiments on differential equations show Randomized Signature finds solutions close to ground-truth solutions. Experiments on time series classification also show its potential.

**Summary Of The Review:**

Using Signature theory for time series analysis is a promising direction with some theoretical guarantees. The paper however is not self-contained and some details are lacking. Please help to address the above comments.

---

> ### Author Response · Authors · 2021-11-18
> **Added runtime comparison, further theoretical insights and details on the activation**
>
> Thanks for your interesting questions, we provide answers below (see new sections in the updated version of the paper).
>
> &nbsp;
>
> - ***Theorem statements are not rigorous.***
>
> A complete formal discussion can be found in Cuchiero et al. (2021a). While the results in this paper are presented in a rather abstract setting, our paper is devoted to a thorough investigation of the practical applicability to time series analysis.
>
> &nbsp;
>
> - ***How to choose activation?***
>
> Thanks for pointing out this deficiency. We devoted a new paragraph called "Note on the choice of $\sigma$ where we point out that we use a linear activation function of slope equal to $1/(d \sqrt{k})$.
>
> &nbsp;
>
> - ***More detailed runtime comparisons between the two types of signature***
>
> This was a common request by reviewers so we dedicated a new Section 4.4 to this discussion, providing compelling experimental results. We show how in the high-dimensional setting Randomized Signatures lead to better performance compared to Truncated Signatures, both in terms of test error, training speed, and memory requirements. We hope this comparison clears your doubts
>
> &nbsp;
>
> - ***Could the authors provide a summary of the 128 UCI data sets for time series classification?***
>
> Added at page 21 under “Dataset Details” paragraph.
>
> &nbsp;
>
> - ***Randomized Signature + Rocket performs worse than Rocket***
>
> As we answered to reviewer 5Jna, our goal is not to beat the SOTA methods for time-series classification and we apologize if that was not clearly explained in the text. Here are a few remarks on your point:
>
> 1) In some sectors (e.g. finance), it is more important to have a method that is theoretically solid with good performance rather than one that performs exceptionally well but has no theoretical motivation (e.g. Rocket, Deep Nets). All in all, our method is fast, sample, efficient, and theoretically sound: all features which make it, in our opinion, better suited for some classes of applications.
>
> 2) RocketOurs is just a very basic first implementation, with no further “tricks”. Furthermore, we note that Rocket was fine-tuned on UCR whereas we did not fine-tune the hyperparameters for our method (sigmoid, k ...).
>
> 3) Our goal is mainly to show that Randomized Signatures, in the same way as Signatures, provide relatively task-agnostic features: they can be used for various tasks (System Identification and Time Series Classification) and incorporated into existing algorithms (Rocket) with minimal effort and with high performance. Differently from Signatures though, Randomized Signatures are cheaper in the regime of high-dimensional and much more effective in small datasets (see new Section 4.4).

---

### Official Review · Reviewer_5Jna · 2021-11-02

**Correctness:** 3
**Technical Novelty And Significance:** 2
**Empirical Novelty And Significance:** 4
**Recommendation:** 5
**Confidence:** 5

**Main Review:**

S1：This paper gives significant insights into the randomized signatures by showcasing its performance on non-parametric system identification and time series classification problems.

S2: The background and theory related to randomized signatures are well described.

S3: The experimental results provide evidence of the effectiveness of the presented strategy.

W1: The original contribution of this paper is limited. The methods of reservoir computing and random signature are mature components. The paper just uses these strategies in a trivial manner.

W2: The claimed advantages of the proposed randomized signature layer over the existing solution, such as inferring mapping from data, are not evident. It is expected that the results are provided to show the success of the randomized signature layer on resolving the problems such as the pointed over-parametrization, data hungriness, expensive training cost.

W3: Although the signature is well theoretically guaranteed, it is not clarified how these guarantees can contribute to the problem of time series classification in comparison with the original Rocket where the randomized convolution kernels are used.

W4: A minor issue. As the main contribution of this paper is to showcase the effectiveness of the randomized signature layer, it’s expected that its performance on more representative tasks are evaluated and reported.


**Summary Of The Paper:**

This paper showcases the effectiveness of a recently introduced reservoir of random features, i.e., randomized signatures.

**Summary Of The Review:**

In summary, I believe this paper is under the acceptance threshold because of its limited contributions as well as some other issues, as I have pointed from W1 to W4.

---

> ### Author Response · Authors · 2021-11-18
> **Advantages of Randomized Signature clarified**
>
> Thanks for your review and for the nice comments on our work, we address your concerns below
>
> &nbsp;
>
>
> - ***The paper just uses these strategies of reservoir computing in a trivial manner***
>
> As far as we know, our method has never been developed before and it is not trivial in light of its deep mathematical background including elements of non-commutative algebra and rough path theory (published only in 2020-2021). We care to point out that, as we mentioned also in the reply to iWAG, the implementation of the ideas in Cuchiero et al. (2021) is not straightforward: the paper is purely theoretical, uses language from a different community, and provides limited insights on applications (no implementation/experiments of any kind are provided). The implementation of these ideas requires a substantial engineering effort and makes the method available to practitioners.
> In addition, the connection we provide to existing machine learning methods such as Rocket is novel and we believe it is very interesting: gives theoretical grounding to the technique, and presents a thought-provoking variation on this method.
>
> &nbsp;
>
> - ***The claimed advantages of the proposed randomized signature are not evident. Randomized signature layer on resolving the problems such as the pointed over-parametrization, data hungriness, expensive training cost.***
>
> The advantages in terms of the number of training parameters are clear (see Section 3.2). To better illustrate this point, we show in our new Section 4.4 how in the ***high-dimensional setting*** Randomized Signatures lead to better performance compared to Truncated Signatures: in terms of test error, training speed, and memory requirements. We also provide a new comparison with NCDEs on page 7. Finally, in comparison with deep learning techniques, the advantages are clear in terms of overparametrization – and the classification results also prove that our (low dimensional) features are able to provide surprisingly good performance. We hope that this clears the reviewer’s doubts.
>
> &nbsp;
>
>
> - ***Although the signature is well theoretically guaranteed, it is not clarified how these guarantees can contribute to the problem of time series classification in comparison with the original Rocket where the randomized convolution kernels are used.***
>
> Our goal is not to beat the state-of-the-art methods for time-series classification and we apologize if that was not clearly explained in the text. Here are a few remarks on your point:
>
> 1) In some sectors (e.g. finance), it is more important to have a method that is theoretically solid with good performance rather than one that performs exceptionally well but has no theoretical motivation (e.g. Rocket, Deep Nets). All in all, our method is fast, sample, efficient, and theoretically sound: all features which make it, in our opinion, better suited for some classes of applications.
>
> 2) RocketOurs is just a very basic first implementation, with no further “tricks”. Furthermore, we note that Rocket was fine-tuned on UCR whereas we did not fine-tune the hyperparameters for our method (sigmoid, k ...).
>
> 3) Our goal is mainly to show that Randomized Signatures, in the same way as Signatures, provide relatively task-agnostic features: they can be used for various tasks (System Identification and Time Series Classification) and incorporated into existing algorithms (Rocket) with minimal effort and with high performance. Differently from Signatures though, Randomized Signatures are cheaper in the regime of high-dimensional and much more effective in small datasets (see new Section 4.4).
>
> &nbsp;
>
> - ***Add performance on more representative tasks***
>
> We would be happy to include additional tasks, could you please be more specific?

---

### Official Review · Reviewer_iWAG · 2021-11-02

**Correctness:** 4
**Technical Novelty And Significance:** 1
**Empirical Novelty And Significance:** 3
**Recommendation:** 6
**Confidence:** 4

**Main Review:**

The manuscript builds on previous work that has showed that the signature is a reservoir by demonstrating the performance of the randomized signature. This adds to the literature on reducing overhead in signature methods, which has grown a lot in the last few years but is still a very active area of research. As it stands however, the theoretical contribution of the manuscript is minimal as far as I can tell, and it is mainly a showcase of what can be done with the randomized signature.

While the breadth of experiments is impressive, there is a lack of comparison with other signature based baselines such as neural CDE or RDE methods or low-rank projections. Ideally there should also be some discussion about what the benefits are of using the randomized signature over the other methods and when it is more or less suitable.

Minor points:

 - Text is, in my opinion, overly verbose and would probably improve if cut down a bit.

 - Many published papers are cited as arXiv preprints.

 - Section 3.2 paragraph 1: "problem, in particular kernelization techniques, see, e.g., Kidger and Lyons (2020)" This paper does not propose kernelization techniques.

Overall, the paper is decent and might be very useful in certain situations, but it lacks discussion, and the empirical contributions are rather weak considering the lack of technical innovation.


**Summary Of The Paper:**

The manuscript proposes that by combining the signature transform with ideas from reservoir computing one may reduce computational cost and spatial overhead. This is demonstrated on some toy problems as well as some real world datasets


**Summary Of The Review:**

This is a good showcase of the power of reservoir computing in signature methods, but there is room for improvement.

---

> ### Author Response · Authors · 2021-11-18
> **Added comparison with other signature-based baselines**
>
> Thanks for your thorough review, we address your concerns below
>
> &nbsp;
>
> - ***Theoretical contribution of the manuscript is minimal***
>
> Our work builds on Cuchiero et al.(2021), which presents the first rigorous discussion of the potential of Randomized Signature. Our contribution is to adapt their deeply mathematically involved findings to the machine learning-oriented community of ICLR. Additionally and more importantly, we empirically validate the conclusions drawn by Cuchiero et al. around the concept of Randomized Signature. We provide evidence that their claims are reflected on several real-world applications, thus showcasing the effectiveness of Randomized Signatures. We believe that our findings are quite surprising since the method is based on random projections: we discuss the connection to existing machine learning literature and show how the method can be applied successfully to real-world scenarios, with stunning performances. We are also not aware of any previous work that discusses the application of the Randomized Signature approach to classification.
> Additionally, we stress that the implementation of the ideas in Cuchiero et al. (2021) is not straightforward: the paper is purely theoretical, uses language from a different community, and provides limited insights on applications (no implementation/experiments of any kind are provided). The implementation of these ideas requires a substantial engineering effort, and we hope our work can be useful for several practical research in many fields (e.g. system identification in biology, finance) – we are going to release the code to the public upon acceptance.
> We also believe that our work leaves several directions to be explored: use of additional tricks (nonlinearities such as saturations), study of the dynamical system entailed by our update rule, connection to Rocket, etc.
>
> &nbsp;
>
> - ***Lack of comparison with other signature-based baselines such as neural CDE or RDE methods***
>
> To address the reviewer's concerns, we present the following experiments in our updated version.
>
> 1)  In Section 4.4 we benchmark the performance of our approach against one leveraging the Truncated Signature of order M, for the task of black-box system identification (predicting solution of ODE given controls). This method takes multi-dimensional controls as input, extracts their Truncated Signature of order M, and maps them to the ODE solutions. We compare this technique to an equivalent one using Randomized Signature instead. The experiment is conducted so that the two techniques share (roughly) the same computational cost necessary to extract the respective features. We are particularly interested in the scenario where the number of input controls is high (>10). We assess the performance in terms of runtime and test error for identification of a Fractional Ornstein–Uhlenbeck process (high roughness). The gain provided by our method is apparent: the number of trainable parameters, as well as the training time of the Truncated Signature-based model, scale exponentially with the number of controls, whilst its test-error performance degenerates. We invite the reader to check Figure 7.
>
> 2) We already provided a comparison of our method with a strong baseline for system identification: please check the comparison with NNARX in Figure 6. However, as the reviewer requested, in our updated version (Figure 13) we benchmark our model also against NCDEs on the task of learning the law of an Ornstein–Uhlenbeck process. We train three versions of such NCDE with three different orders of magnitude of trainable parameters and report test performance. All three models obtain errors that are 4 orders of magnitude larger than our model. Additionally, the training time is at least 40k times bigger than our model (we used the official code). This experiment, along with our qualitative comparison with NNARX allows us to conclude that: a) Randomized Signatures performs very well on non-parametric system identification; b) they are advantageous since they work well with little amounts of data and involve a very small number of trainable parameters (readout). This is explained by Cuchiero et al., which clearly states that Randomized Signature inherits the universal approximation property from Signature and thus enjoys a strong inductive bias for this task.
>
> &nbsp;
>
> - ***Benefits are of using the randomized signature over the other methods***
>
> Our advantages are apparent in the regime of high-dimensional datasets with few data points, which are of great practical interest in finance and industry. Please also check the new Sec. 4.4 and the new paragraph on NCDEs on page 7.
>
> &nbsp;
>
> - ***Minor points***
>
> Thanks for this useful feedback, we corrected the citations from arXiv to journals/conferences and updated the sentence in Section 3.2. For a potential camera-ready, we will shorten the text and include in the paper the complete NCDE experiment (now in the Appendix).

---

### Author Response · Authors · 2021-11-22
**Requested experiments done**

Dear all, thanks a lot for the time in reading our rebuttal. Here is a quick summary (sort of a TL;DR) of how we addressed the main reviewers' concerns.
We are happy about this feedback; it helped us better showcase our contributions. Furthermore, we are positive that the additional experiments we performed will positively surprise the reviewers.

&nbsp;

***Concern*** :  Lack of (runtime) comparison with other signature-based baselines.

***Modification*** : We added a subsection (Section 4.4) where we show experimentally how, in the high-dimensional setting (many controls), Randomized Signatures _lead to better performance than Truncated Signatures – in terms of test error, training speed, and memory requirements_. We also provide a new comparison with Neural Controlled Differential Equations (Kidger et al. 2020) on page 7, showing again superiority of randomized signatures in system identification. Finally, we note that we used the publicly available code for all the methods we compared against.

&nbsp;

***Concern*** : Background theorems too informal + more intuition needed

***Modification*** : We added a citation to the paper where the formal statements can be found (Cuchiero et al. 2021a). We modified a few lines in Theorem 8 (appendix) to make the stated result more formal. To enhance intuition and give an idea of the proof, we added a few lines after Theorem 2.

&nbsp;

***Concern*** : Theoretical contribution of the manuscript is minimal

***Answer*** : Our work builds on Cuchiero et al.(2021a,b), which present the first rigorous discussion of the potential of Randomized Signature. Our contribution is to adapt their deeply mathematically involved findings to the machine learning-oriented community of ICLR. The implementation of the ideas in Cuchiero et al. (2021) is not straightforward: the paper is purely theoretical, uses language from a different community, and provides limited insights on applications (no implementation/experiments of any kind are provided). The implementation of these ideas requires a substantial engineering effort, and we are positive our work can be helpful in several practical research in many fields (e.g., system identification in biology, finance).
At a higher level, our work provides evidence that claims of _theoretical benefits of randomized signatures are reflected on several real-world applications_, thus showcasing the effectiveness of Randomized Signatures. We believe that our findings are pretty surprising since the method is based on random projections: we discuss the intriguing connection to existing machine learning literature (not done in Cuchiero et al.) and show how the method can be applied successfully to real-world scenarios, with stunning performances and little computational costs in comparison to other signature-based approaches and deep learning approaches.

---

### Comment · Reviewer_5Jna · 2021-11-28
**Feedback to the discussions**

For W2, it’ is expected to show the proposed advantages clearly, both in theoretical analysis and in experiment evaluation.

For W3, as the authors claimed, the randomized signature is better than ROCKET because it is theoretically guaranteed. However, what the associated theory can benefit the TSC problem is not clarified (Maybe it contributes to the interpretability but we cannot see in the paper).  Straightforward evidence is expected.

For W1 and W4, I believe the contribution between theory and showcase is a tradeoff when evaluating this paper. As other reviewers have also pointed out, the theoretical contribution of this paper is minimal. The authors have also said in your reply that your contribution focuses on the implementation of the original ideas and the efforts are paid on engineering issues. I believe these efforts are not academic contributions. So I prefer the contributions of this paper in the interesting showcases. That is why I propose W4. And I expect the authors to showcase more feature-based tasks for time-series, such as time-series prediction and outlier detection.

---

> ### Author Response · Authors · 2021-11-30
> **Reply to your concerns**
>
> We thank the reviewer for their feedback
>
> **W2 :  it is expected to show the proposed advantages clearly**
>
> The advantages of Randomized Signature (RS) compared to Signature are clearly shown in Fig. 7. Theoretical considerations are provided in Section 3.2.
>
> In particular, looking at Fig. 7, it is apparent that RS yields significant advantages in terms of (a) the number of trainable parameters, (b) training time, (c) performance in the regime of small datasets (when N is small) and high dimensional controls (when d is large). We note that this experiment is performed under a highly controlled setting, and the findings are well supported by theoretical considerations on the underlying optimization problem (see Section 4.4 for details).
>
> Additionally, we also provide clear evidence that our method outperforms two deep learning baselines,  namely NNARX (a strong popular baseline for non-parametric system identification) and NeuralCDE (as recommended by Reviewer iWAG) in two different challenging system identification tasks.
>
> _We kindly invite the reader to take a closer look at these sections – thank you._
>
> **W3: how RS can benefit the TSC problem is not clarified**
>
> The goal of the TSC section was not to show SOTA results in time series classification but simply to strengthen our claim that the RS can be used as a powerful reservoir. This is stated clearly in our contribution section.
>
> Nevertheless, our experiments show that the performance is very close to a State-Of-The-Art model like Rocket and that it is superior to various deep learning baselines (which are notoriously data-hungry and tricky to train). _We believe this is not a trivial finding_, especially in light of the very little fine-tuning we did on our model – as opposed to Rocket which is accurately tuned to the time series datasets we considered.
>
> That being said, we believe our experimental evaluation is quite complete we show the performance of our method in
>
> - System identification: toy settings (linear system, irregularly sampled time grid, fractional noise) + 3 highly non-linear real-world applications (2 biological, 1 chemical). We compare to other approaches and discuss advantages compared to: signature methods, deep learning methods, and classical methods (such as NNARX).
>
> - Classification: We run randomized signatures on the complete UCR archive – which consists of 128 datasets. We believe is a reasonably large benchmark to demonstrate that RS provides a powerful reservoir.
>
> **W1 and W4:  I expect the authors to showcase more feature-based tasks for time-series**
>
> The idea of our paper is to showcase the expressiveness of RS in several challenging tasks involving time series. We did so by considering 8 system identification tasks (from biology, engineering, and medicine) and 128 TSC datasets, a comprehensive breadth of experiments judged as impressive by reviewer iWAG. Our performances are excellent in both system identification and TSC, thus providing strong empirical evidence that RS represents a powerful reservoir. We further stress that these results have never been shown before, and Cuchiero et al. do not provide any experimental assessment of their results.
>
> We also kindly note that our experimental investigation is perfectly in line with numerous published papers on Signature-based algorithms (e.g. Bonnier et al, 2019, Kidger et al 2020), and shows clear advantages compared to these methods.
> Further, we are not aware of any published paper on signature-based methods that discuss problems such as outlier detection.
>
> In essence: _we believe our experiments are compelling_, also in light of the related literature. Yet, we would have been delighted to satisfy the requests of reviewer 5Jna to do additional experiments on "more representative tasks." However, unfortunately, the reviewer was not specific enough in the request (as we noted in the rebuttal).

---

### Decision · Program_Chairs · 2022-01-20

**Decision:**

Reject

**Comment:**

This paper empirically evaluates the performance (in time and accuracy) of randomized signatures for time series, an idea that was developed theoretically in a series of recent paper. While reviewers acknowledge that implementing and testing this idea is relevant, they also consider that the lack of methodological and theoretical novelty, combined with the fact that the experimental results do not convincingly show that randomized signatures outperform existing methods on a variety of tasks, puts the paper below the acceptance bar.